

# Post Little Ice Age rock wall permafrost evolution in Norway

Justyna Czekirda[1,*], Bernd Etzelmüller[1], Sebastian Westermann[1], Ketil Isaksen[2], Florence
Magnin[3]

[1]Department of Geosciences, University of Oslo, 0316 Oslo, Norway
[2]Department of Research and Development, Norwegian Meteorological Institute, 0313 Oslo, Norway
[3]EDYTEM, Université Savoie Mont-Blanc, CNRS, 73000 Chambery, France

*Correspondence to*: Justyna Czekirda (justyna.czekirda@geo.uio.no)

**Keywords**: *Rock walls, Mountain Permafrost, Norway, Thermal modelling, Permafrost modelling, Climate change*

**Abstract**

Around 10 % of unstable rock slopes in Norway are possibly underlain by widespread permafrost. Permafrost thaw and degradation may play a role in slope destabilization and more knowledge about rock wall permafrost in Norway is needed to investigate possible links between ground thermal regime, geomorphological activity and

natural hazards. Here, we assess spatio-temporal permafrost variations in selected rock walls in Norway over the last 120 years. We model ground temperature using the two-dimensional ground heat flux model CryoGrid 2D along nine profiles crossing monitored rock walls in Norway. The simulation results show the distribution of sporadic to continuous permafrost along the modelled profiles. Ground temperature at 20 m depth in steep rock faces increased by 0.2 °C decade$^{-1}$ on average since the 1980s. Rates of ground temperature change increase with

elevation within a single rock wall section. Multi-dimensional thermal effects are in general smaller in Norway than in e.g. the European Alps due to gentler mountain topography and less aspect-related variations in ground surface temperature. Nevertheless, the steepest mountains are still sensitive to even small differences in ground surface temperature. This study further demonstrates how rock wall permafrost distribution and/or rock wall temperature increase rates are influenced by factors such as surface air temperature uncertainties, surface offsets

arising from the incoming shortwave solar radiation, snow conditions in, above and below rock walls, rock wall geometry and size, adjacent blockfield-covered plateaus or glaciers.

## 1 Introduction

Numerous studies infer that thawing permafrost induced rapid mass movement events around the world, e.g. in the European Alps (Dramis et al., 1995; Fischer et al., 2006; Ravanel et al., 2010), the New Zealand Southern Alps

(Allen et al., 2009), Alaska (Huggel et al., 2010) and the Caucasus (Haeberli et al., 2004). Concerns arise because inventories from the European Alps document an enhanced frequency of rockfalls and rock avalanches from permafrost rock walls, especially at the lower permafrost limit, since around 1990/2000 in response to accelerated global warming (Ravanel and Deline, 2011; Fischer et al., 2012). An example of a fast response was exceptional rockfall activity that was reported during the hot summers of 2003 and 2015 in the European Alps, likely because

of permafrost degradation (Gruber et al., 2004; Ravanel et al., 2017). Deep permafrost requires longer timescales to degrade and its warming or degradation likely resulted in the activation of the slowly creeping rock masses in the warmer period of the Holocene Thermal Maximum thousands of years after local deglaciation (Lebrouc et al.,



2013; Böhme et al., 2019; Hilger et al., 2021). The stability of permafrost-underlain cliffs with the consequent hazards, such as rockfalls or rock avalanches, is of growing concern considering global surface warming projections. Rock wall permafrost is highly susceptible to climate deterioration because: (1) ice contents are typically low in bedrock causing a quicker ground temperature (GT) increase (Gruber and Haeberli, 2007), (2) surface temperature change enters several mountainsides and results in faster degradation of deeper permafrost in some locations than would be the case in flatter terrain (Noetzli et al., 2007), and (3) thermal conditions in steep bedrock and atmosphere are strongly coupled since steep slopes are typically covered with shallow snow, debris or soil if any (e.g. Boeckli et al., 2012; Myhra et al., 2017).

Several authors have linked permafrost degradation and destabilisation of slopes (e.g. Davies et al., 2000; Davies et al., 2001; Gruber and Haeberli, 2007; Krautblatter et al., 2013). Conductive warming of ice-filled fractures, which are believed to stabilise permafrost-underlain mountains (e.g. Dramis et al., 1995), may result in: (1) loss of joint bonding and reduction of shear strength of the joint due to water release through ice melting, (2) shear strength changes due to mechanical ice properties that are a function of the normal stress and temperature (Davies et al., 2001). Furthermore, advective heat transport by percolating meltwater might result in rapid, local degradation of rock wall permafrost, which can trigger rockfalls even in cold permafrost areas (Hasler et al., 2011b). Krautblatter et al. (2013) noticed, in addition, that rock-mechanical properties themselves depend on rock temperature; hence, thawing can lead to a significant drop in rock strength. Frost weathering processes caused by ice segregation and/or volumetric expansion are believed to contribute to the generation of weakness planes or widening fractures in frost-affected rocks (Gruber and Haeberli, 2007; Krautblatter et al., 2013).

Some impacts of permafrost degradation on the dynamics of recent rock-slope failures are likely for a few sites in Norway, e.g. the Gámanjunni-3 instability in the Northern Norway that accelerated recently (Böhme et al., 2019), the Polvartinden rock avalanche in the Northern Norway that occurred in 2008 (Frauenfelder et al., 2018) or possibly for the north-facing Veslemannen in southern Norway that fell in 2019, where at least seasonal freezing controlled the rock stability (Kristensen et al., 2021). Moreover, Blikra et al. (2006) discussed permafrost thawing as a possible triggering mechanism for rock-slope failures that occurred after the deglaciation in Norway. Hilger et al. (2021) modelled permafrost distribution in the Holocene and suggested that permafrost had likely a stabilising effect on some rock slopes in Norway for several millennia after deglaciation. Magnin et al. (2019) estimated that 11 % of potentially unstable slopes in Norway are currently underlain by at least discontinuous permafrost.

Numerous studies concerning permafrost in the gentle parts of the Scandinavian Mountains have been published since the 1980s, attributing variations in mountain permafrost occurrence owing to mean annual air temperature (Etzelmüller et al., 1998), elevation (Sollid et al., 2003; Heggem et al., 2005), snow cover (Farbrot et al., 2008; Farbrot et al., 2011; Isaksen et al., 2011; Gisnås et al., 2017), blockfield cover or surficial sediments (Farbrot et al., 2011; Gisnås et al., 2017), and vegetation cover (Farbrot et al., 2013; Gisnås et al., 2017). Studies indicate that recent atmospheric warming has led to the degradation of mountain permafrost in gentle terrain in Norway, especially since the 1990s (Isaksen et al., 2007; Hipp et al., 2012; Westermann et al., 2013; Etzelmüller et al., 2020).

The earliest rock wall permafrost studies in Norway provided: 1) first rock wall temperature measurements from rock faces in the Jotunheimen, central-southern Norway (Hipp et al., 2014), and from small rock cliffs in Troms, the Northern Norway (Frauenfelder et al., 2018), 2) first-order rock wall permafrost map for mainland





Norway based on the statistical permafrost model relating permafrost distribution to both elevation and potential incoming short-wave radiation (Steiger et al., 2016), and 3) first 2D modelling for three north-facing rock walls in Norway, based on the interpolated air temperature, variable snow cover and presence of glaciers (Myhra et al., 2017). Systematic field observations using rock wall loggers that were installed at selected sites in Norway in 2010 in the Jotunheimen (Hipp et al., 2014) and from 2015 through 2017 at other sites across southern and northern Norway (Magnin et al., 2019) allowed later for the improvement of the earlier approaches of Hipp et al. (2014) and Steiger et al. (2016) to calibrate a near-surface thermal regime model for rock wall permafrost in Norway by using mean annual air temperature (MAAT) as an explanatory variable instead of elevation. In this study, we employ the 2D slope-scale transient heat flow model CryoGrid 2D (Myhra et al., 2017) to simulate the thermal state of mountain permafrost along transects crossing the instrumented rock walls in mainland Norway since 1900. We advance the methods presented in the study by Myhra et al. (2017), by the observation-constrained modelling, i.e. including the field observations from rock walls in various expositions. As presented in Magnin et al. (2019) lower elevation limits of near-surface permafrost vary between southern- and northern-facing slopes in Norway by several hundred metres, even though the elevation difference is less pronounced than e.g. in the European Alps. We included nine transects in the present study to improve the knowledge of permafrost geometries and over-a-century-long permafrost development in steep rock faces across Norway. These results are a prerequisite for stability assessment in the Norwegian rock walls.

## 2 Study areas and field installations

### 2.1 Western Norway

The Western Norway is characterised by alpine mountains, deep glacial valleys and fjords, which were formed after multiple mountain and full-sized Fennoscandian ice sheets linearly eroded the pre-existing fluvially eroded valleys (Kleman et al., 2008). The climate in the area is maritime with annual precipitation sums of more than 2000 mm (Lussana, 2018) and an annual range of mean temperature of less than 18 °C (Tveito et al., 2000). Permafrost limit is higher in this part of Norway as high-elevation areas are often occupied by glaciers or deeper winter snow, which isolates the ground in many places (Etzelmüller et al., 2003). The areas where permafrost research was conducted include sporadic permafrost at Finse at the Hardangervidda Mountain Plateau (Gisnås et al., 2014) and during 2015–2017 nine rock wall loggers have been installed at selected sites in the Western Norway (Magnin et al., 2019). Lower rock wall permafrost limits in the area can be at present expected at 1300–1400 m elevation in north-facing slopes (Magnin et al., 2019).

We choose four profiles in the Western Norway for this study: (1) Mannen (Figure 1G), (2) Hogrenningsnibba (Figure 1B), (3) Kvernhusfjellet (Figure 1B) and (4) Ramnanosi (Figure 1C). The name Mannen is used for both a mountain peak at 1294 m elevation and a large active rockslide in the Møre og Romsdal county. Over the last few years, it has been moving with a velocity of more than 20 mm a⁻¹ in the upper part of the slope above about 1000 m elevation (Etzelmüller et al., 2021). The Mannen instability activated during the Holocene Thermal Maximum around 8 ka (Hilger et al., 2021), leading to the formation of a 20 m high backscarp. The rockslide developed in the Caledonian metamorphic rocks of the so-called Western Gneiss Region, within a quartz-rich gneiss unit with sillimanite and kyanite minerals (Saintot et al., 2012). Hogrenningsnibba (1670 m) and Kvernhusfjellet (1740 m) are two mountains located above the Raudalen Valley north of the Jostedalsbreen Ice





Cap, on the eastern side of Lovatnet Lake and next to the Tindefjellbreen Glacier. On the other side of Lovatnet
Lake, two of the worst natural disasters in Norwegian history occurred in 1905 and 1936 when rock avalanches
from Ramnefjell Mountain ("Loenulykkene") generated tsunami waves and killed many people living in the
valley. Bedrock at Hogrenningsnibba is mapped as quartz monzonite, whereas Kvernhusfjellet is mapped as
composed mainly of granitic gneiss (Lutro and Tveten, 1996). Ramnanosi (1421 m) is a mountain peak in the Flåm
Valley in the former county of Sogn og Fjordane. Ramnanosi peak is part of the larger unstable rock slope Stampa,
which includes the continuously monitored Joasetbergi instability around 3 km north of Ramnanosi. Around the
Ramnanosi Mountain, both gravitational faults and fractures were mapped in the phyllite nappes. Below a west-
facing 200 m high slide scar there are deposits from the rock avalanche/rockfall events (Blikra et al., 2006; Böhme
et al., 2012; Böhme et al., 2013).

**2.2 Jotunheimen**

The Jotunheimen Mountain Range is located in the central part of southern Norway and represents one of the
highest mountain areas in Norway, including the highest peak in Norway, Galdhøpiggen (2469 m). Till deposits
are more common in the Jotunheimen, where glacial erosion has been small compared with the Western Norway
due to its proximity to the ice divide (Olsen et al., 2013). At high-mountain plateaus, blockfields were preserved
beneath non-erosive ice sheets (Sollid and Sørbel, 1994; Goehring et al., 2008), where negative thermal anomaly
in blockfields could enhance the formation of permafrost and cold basal conditions (Juliussen and Humlum, 2007).
The Jotunheimen area receives less precipitation than the Western Norway with mean precipitation typically less
than 1000 mm per year in the normal period 1961–1990 (Lussana, 2018) and has an annual range of mean
temperature of normally greater than 18 ℃ (Tveito et al., 2000). Most mountain permafrost research in southern
Norway has been conducted in the Central and Eastern Norway, especially in the Jotunheimen Mountain Range
(Ødegård et al., 1992; Farbrot et al., 2011; Isaksen et al., 2011). In 1982, the first 10 m deep borehole at 1851 m
elevation was drilled in the Jotunheimen (Ødegård et al., 1992) and then in August 1999, the deepest permafrost
borehole (129 m) in Norway was drilled in the continuous permafrost zone at Juvvasshøe (1894 m) as part of the
PACE project (Sollid et al., 2000; Harris et al., 2001). Six additional boreholes have been drilled at various
elevations in the Juvvasshøe area on its north-eastern slope in August 2008 (Farbrot et al., 2011). The measured
GTs show that discontinuous permafrost occurs down to at least the borehole drilled in the bedrock at 1559 m
elevation (Juv-BH4). For the second lowest borehole (Juv-BH5) and nearby gentle slopes, geophysical surveys
performed in 1999 to delineate the elevation limit of mountain permafrost were repeated in 2009 and 2010 and
indicated the degradation of permafrost over the intervening decade (Isaksen et al. 2011). At the highest elevations
(above ~1850 m elevation) permafrost has likely been present throughout the whole Holocene (Lilleøren et al.,
2012).  Magnin et al. (2019)'s statistical model results suggested that the lower limit of rock wall permafrost in
the Jotunheimen area is at approximately 1550 and 1150 m elevation in the south- and north-facing rock walls,
respectively.

We define two profiles in the Jotunheimen in this study (Figure 1D) with (1) Veslpiggen (2369 m) and (2)
Galdhøe (2283 m). The selected profiles are mostly within the tectonic unit of the Jotun-Valdres Nappe Complex
and the bedrock along the profiles is composed of pyroxene granulite (Lutro and Tveten, 2012).



### 2.3 Northern Norway

The geomorphology of Northern Norway is in general similar to southern Norway with multiple glaciations leading to the formation of fjords and U-valleys and the depositional areas further inland, e.g. at Finnmarksvidda (Kleman et al., 2008; Olsen et al., 2013). The climate in the Northern Norway is mostly subarctic in the lowland and tundra type in the mountains. The climate varies from maritime in the coastal areas, with the largest precipitation sums in the Nordland county > 2000 mm in 1961–1990 (Lussana, 2018), to a more continental character at Finnmarksvidda, where annual precipitation sums were less than 750 mm in 1961–1990 (Lussana, 2018). Several permafrost studies have been conducted in the Northern Norway, where both miniature temperature dataloggers and borehole temperature strings were installed to monitor ground thermal conditions (Isaksen et al., 2008; Christiansen et al., 2010; Farbrot et al., 2013). Farbrot et al. (2013) distinguished three permafrost regions in the Northern Norway (excluding Nordland): (1) maritime mountain permafrost in the western part of Troms county, (2) continental permafrost in Finnmark, mainly in palsa mires, and (3) Low Arctic permafrost at the Varanger Peninsula. For the gentle terrain, permafrost limits decrease from 800–900 m elevation in the western areas of Troms to around 200–300 m elevation in the continental parts of Finnmark and Troms (Farbrot et al., 2013).

Three transects in the coastal areas of the Northern Norway are extracted in this study: (1) Gámanjunni 3 (Figure 1A), (2) Ádjit (Figure 1E), (3) Rombakstøtta (Figure 1F). Gámanjunni 3 (Figure 1A) in the Manndalen Valley, west of Tromsø, is one of the most unstable rock slopes in Norway, moving recently up to 60 mm a$^{-1}$ (Böhme et al., 2016a; Böhme et al., 2019; Etzelmüller et al., 2021). The unstable part has moved approximately 150 m down since the end of the Holocene Thermal Maximum (Böhme et al., 2019; Hilger et al., 2021). The upper part of the profile (>300–500 m elevation) is composed of mica schists that are part of the Kåfjord Nappe, whereas the lower parts of the profile are mapped as calcareous mica-schist within the Váddás Nappe and superficial deposits of Quaternary age (Quenardel and Zwaan, 2008). Ádjit (Figure 1E) is a mountain ridge in the Skibotn Valley, Troms, where below its south-western rock wall several periglacial and mass movement landforms were mapped, such as e.g. active and inactive talus-derived rock glaciers (Nopper, 2015; Eriksen et al., 2018). The mountain is located within the Kåfjord Nappe and meta-arkose to feldspathic quartzite, together with mica or garnet-mica schists dominate along the profile (Boyd et al., 1985; Quenardel and Zwaan, 2008). Rombakstøtta (Figure 1F) is a steep mountain top at 1230 m elevation located a few kilometres east of Narvik, Nordland. Geologically, Rombakstøtta profile is within the Narvik Nappe Complex, and the rock types mapped along the profile are garnet or kyanite-garnet two-mica schist, quartzite and quartzite schist (Karlsen, 1991). The north-facing part of the mountain, east of our profile, displays open tension cracks and it has been subject to investigations due to its instability potential (Gauer et al., 2016; Morken, 2017).

## 3 Methods

### 3.1 CryoGrid 2D

A transient 2D heat conduction model, CryoGrid 2D (Myhra et al., 2017), is employed to model GT evolution along the selected profiles. The subsurface temperature is modelled by solving the heat diffusion equation following Fourier's law of heat conduction with the material- and temperature-dependent thermal parameters. The effective volumetric heat capacity, which includes the latent heat effects due to water/ice phase transitions, and the thermal conductivity are functions of volumetric contents of soil/rock components (mineral, water/ice, air, organic)



and their individual thermal properties, as defined in the one-dimensional CryoGrid 2 model (Westermann et al., 2013). In CryoGrid 2D, MATLAB-based finite element solver MILAMIN package (Dabrowski et al., 2008) generates an unstructured triangular mesh for a given slope geometry and is used for space discretisation, whereas time discretisation is based on the finite-difference backward Euler scheme. The spatial resolution in the CryoGrid 2D is prescribed by the maximum triangle area (MTA), i.e. a maximum area for the three node triangular elements.

Dirichlet boundary conditions are used at the upper model boundary and the model is forced by GST at the air-ground interface, i.e. temperature below the snowpack. A more thorough description of the model and equations can be found in Myhra et al. (2017). Note that CryoGrid 2D is a conductive model, hence, convective or advective heat transport is unaccounted for. The model is constructed as a 2D slice through a slope, assuming translational symmetry along the third dimension.

**3.2 Model geometry and ground stratigraphy**

The upper boundary for the selected profiles was extracted from the 0.5–1 m digital elevation models (DEMs) available from the Norwegian Mapping Authority at www.hoydedata.no, whereas the lower boundary extends down to 6000 m below sea level. Most profiles are approximately 2.5–4 km long, except for the ~7.5 km long profiles in the Jotunheimen (Figure 2). Because profiles in the Jotunheimen, together with the profile at

Kvernhusfjellet traverse glaciers, we compute glacier bed elevation by extracting glacier thickness provided by the Norwegian Water Resources and Energy Directorate (NVE), where ice thickness was estimated using a distributed model described in detail in Andreassen et al. (2015). At Kvernhusfjellet, we add a 5 m thick snow patch on the top plateau as observed on the orthophotos from the Norwegian Public Roads Administration, the Norwegian Institute of Bioeconomy Research and the Norwegian Mapping Authority (www.norgeibilder.no).

Meshes for each profile are constructed with nodes at a 0.05 m distance at the upper boundary and MTA that increases with depth. The constructed meshes have MTA of 0.05 $m^2$ between the ground surface and 2 m depth, 0.20 $m^2$ at depths between 2 and 10 m, 0.50 $m^2$ at depths between 10 and 20 m, 5.00 $m^2$ at depths between 20 and 100 m, and 50 $m^2$ below 100 m depth. The model domains consist of approximately 500,000 vertices, except for the longer profiles in the Jotunheimen, where each mesh has ~1,250,000 nodes. No mechanical aspect is considered

in this study; hence, the meshes remain static throughout the whole simulation period.

A digital map of superficial deposits is available for the entire Norway from the Geological Survey of Norway (NVE) at 1:250.00 scale. Due to the small scale of the map, we refine the geomorphological mapping along the upper profile boundaries based on the available orthophotos from www.norgeibilder.no. The ground composition (Table 1) is based on the sediments mapped on the surface for most profiles, where we define hard

vertical boundaries between the sediment classes also at depth because such an approach allows for an effective and almost automated generation of nodes for an unstructured mesh. Similar volumetric contents and layers for the NVE sediment classes are assumed as in Westermann et al. (2013) for the one-dimensional CryoGrid 2. However, we apply a higher rock porosity than Westermann et al. (2013) and follow the higher porosity of 5 % vol. to account for rock discontinuities as Myhra et al. (2017). The thermal conductivity for the mineral fraction is

extracted from the same data as in Westermann et al. (2013) and varies for the sites between 2.3 and 3.1 W m $^{-1}$ K$^{-1}$ (Table A1). For the Jotunheimen profiles, we used a value of 2.7 W m $^{-1}$ K$^{-1}$ (Hipp et al., 2012). The NVE sediment classes and their stratigraphy as defined in Westermann et al. (2013) lack a suitable representation for some sediments mapped along the profiles. Therefore, we added a few sediment classes to fill this gap (Table 1).



The Ádjit profile intersects a rock glacier at lower elevations, where we used a similar geometry, as presented in
Eriksen et al. (2018). For Gámanjunni we use a slightly modified version of a geological profile for the unstable
part (Böhme et al., 2016b), in conjunction with the geomorphological mapping outside of the geological model.
The scree class is defined with the same parameters as in Myhra et al. (2019). At Ramnanosi, very thick 30 m thick
colluvium deposits are assumed just below the rock wall down to around 600 m elevation and 4 m thick regolith
is assumed at the plateau. Bedrock stratigraphy is assumed to be below glaciers and perennial snow.

### 3.3 Model forcing

### 3.3.1 Surface air temperature

The modelled daily surface air temperature (SAT) data set for the mainland Norway, hereafter seNorge, is available
for 1 km$^2$ grid cells for the period 1957–present (Lussana, 2020). However, the seNorge data set overestimates
SAT trends and often shows increasing SAT trends with elevation for our study sites, leading to e.g. 3 °C SAT
increase in the Jotunheimen between the 1980s and the 2010s. This is the result of the inhomogeneity in the
network of meteorological stations, particularly the lack of meteorological stations at mountain plateaus in some
periods. Cold periods are overestimated if the gridded data set is based mainly on meteorological stations in
valleys, where air inversions are frequent during winter. Therefore, we choose to force the model with the regional
monthly data set provided by the Norwegian Meteorological Institute, described in detail in Hanssen-Bauer et al.
(2006). This regional model yields robust temporal estimates at a regional scale; nevertheless, rather poor spatial
coverage at local scales. Therefore, we superimpose a local component on the regional data. Regional SAT data
sets were provided for valleys at the bottom of each profile.  We use the following procedure for each data set:

(1) Since we want to begin to run the model at the end of the Little Ice Age (LIA) in Norway and the regional
     SAT data sets start in 1900, we reconstruct SAT back in time by using SAT from the long-term
     meteorological stations described in Table 2. We account for average offsets in the overlapping period
     between SAT from the long-term meteorological stations and the regional SAT.

(2) We adjust regional SATs by subtracting offsets between the regional and local SATs from a nearby
     meteorological station or seNorge for valleys over the last few years.

(3) We compute the average monthly lapse rate between two meteorological stations, typically one at the
     bottom of the valley and one at or close to the mountain plateau over the last few years. The selected SAT
     data are listed in Table 2.

(4) We compute monthly temperature at the mountain top using the monthly lapse rates.

The selected last few years used in this analysis are periods when temperature measurements in the rock walls are
available. This allows for a comparison of SAT with GST in the rock wall loggers and gives more reliability. The
aforementioned procedure allows for the reproduction of similar SAT trends at mountain plateaus as provided for
valleys, hence removing the elevation dependency in the SAT trends present in the seNorge data. We describe 10
year running mean surface air temperature (SAT10a) evolution for the highest elevations along each profile in
Appendix B.



### 3.3.2 Nival offsets

We lack observations of snow cover dynamics and snow depths from the rock walls in Norway. In this study, we are mostly interested in the thermal insulation effect of snow cover and not snow depth itself, especially because our permafrost model lacks an explicit snow domain. In equilibrium permafrost models such as the TTOP-model (Smith and Riseborough, 2002), insulating snow effects are usually accounted for by using nF-factors that link SATs and GSTs. We follow an easy-to-implement hypothesis that snow thickness and its insulating effect on the

GST depend on the slope gradient. Hence, we assign various nF-factors along the profiles according to the computed slope gradient; however, some sediment/vegetation cover types have distinct values for nF (Table 3). We assume that steep slopes, i.e. steeper than 60º are snow-free (discussed in Sect. 5.1.4). Furthermore, we detect 1 m deep sinks along the profiles using fillsinks from TopoToolbox 2 (Schwanghart and Scherler, 2014) and assume that these are areas where snow might accumulate and use the same nF as for the gentlest gradient (slope

$< 30°$) along profiles. Additionally, we assign a special nF value of 0.25, as computed by Gisnås et al. (2017), for the broad-leaved forest (code 311) based on CORINE land cover 2018 (Aune-Lundberg and Strand, 2010). The broad-leaved forest occurs at lower elevations along the profiles.

For the top block at Gámanjunni (slope gradient $< 30°$), we compute nF=0.5 based on the SAT and GST measurements conducted by Eriksen (2018b). For the rock glacier at Ádjit, we found an nF value of 0.8 (Eriksen,

2018a). Measurements from the three uppermost boreholes BH-1 (nF=0.78 in 2008–2019), PACE (nF=0.89 in 1999–2018) and BH-2 (nF=0.37 in 2008–2019) in the Jotunheimen yield an average rounded nF value of 0.70 that we apply for the blockfield locations. We note that nF for the blocky terrain (blockfields and rock glaciers) is not necessarily due to nival offsets and is rather caused by air convection (discussed in Sect. 5.1.1.).

### 3.3.3 Surface offsets

Our analysis of the measured 2 h rock wall temperature indicates that rock wall temperature in Norway is influenced by solar radiation as early as February in the Northern Norway and in all months of the year in southern Norway. Because of their steep vertical slopes, incoming shortwave solar radiation might not necessarily be the largest during June, as expected for a horizontal surface at the latitudes in Norway. In the case of rock walls, nT-factors (Smith and Riseborough, 2002) might thus not be able to account for surface offsets (SOs) due to the

shortwave solar radiation in the months when solar radiation is maximum and SAT is still negative, which might occur in the spring months. Additionally, reflected solar radiation from the surrounding terrain is likely an important factor during spring/early summer when snow cover might be present or during the whole year in the rock walls above glaciers. Instead of using temperature transfer factors, we add measured average monthly SOs to SATs at the location of rock walls along profiles. Measured monthly SOs are computed as a difference between

monthly mean ground surface ($GST_{month}$) and surface air ($SAT_{month}$) temperature:

$$SO_{month} = GST_{month} - SAT_{month}. \tag{1}$$

Note that we refer to both rock surface and soil surface temperatures as GSTs in this study. We apply the same SOs to all steep parts of slopes ($>60°$) along profiles and to all months during the entire modelling period. Table 4 summarises the aspects along profiles and selected rock wall loggers to account for the monthly SOs. In this study, SOs is usually referred to SOs arising mainly from solar radiation, unless other indicated.



### 3.4 Model initialisation, model runs and sensitivity tests

We start to run the model around the end of LIA in Norway when the long-term SAT data from meteorological stations are available for correlation (1861/1864 for the profiles in southern Norway, 1874 for the profiles in the Northern Norway). CryoGrid 2D is initialised in a two-step procedure: (1) by running a steady-state version of the model using the average GST for the first decade of the available data and the geothermal heat flux at the lower boundary, (2) spin-up of the model at monthly time steps around 50 times, which yields temperature difference between the consecutive runs at the order of $10^{-4}$ °C. After this initialisation procedure, we continue to run the model at monthly time steps. Accounting for additional at least 20 years of initialisation period, we present the results of the model runs since 1900. Zero heat flux condition is assumed along the vertical left and right boundaries. An average value of geothermal heat flux of 50 mW m$^{-2}$ (Slagstad et al., 2009) is applied at the lower boundary at all sites, except for the profiles in the Jotunheimen, where a value of 33 mW m$^{-2}$ is used (Isaksen et al., 2001). Beneath modern glaciers or perennial snow, we apply GST of 0 °C corresponding to the temperate bed conditions, except for the shallower glaciers or ice patches along the Galdhøe profile in the Jotunheimen, where we apply cold basal conditions at -3 °C as measured in the Juvfonne ice patch (Ødegård et al., 2017). We note, however, that the assumed temperate bed conditions should be rather represented by polythermal bed conditions because the thinnest parts of glaciers have likely temperatures below the pressure melting point (Etzelmüller and Hagen, 2005).

We conduct model sensitivity for all profiles by rerunning the model, including the initialisation steps, for several scenarios. However, we note that some runs are to test the uncertainty in the runs, and some are control runs to investigate the thermal influence of e.g. glacier cover, sediments or SOs in the rock walls. Sensitivity scenarios are listed in Table 5.

## 4 Results

### 4.1 Surface offsets

Figure 3 shows the monthly SOs for the majority of rock wall loggers in Norway. The south-facing loggers usually have the maximum monthly SOs in April, whereas the rest of the loggers often have the maximum monthly SOs in May. There are a few exceptions, e.g. rock wall loggers at Mannen and Rombakstøtta indicate the maximum monthly offsets solely in June. The observation-constrained modelling yields zero mean error and an RMSE below 1.40 °C for the monthly GSTs and significantly improves the correlation between the forcing data and the rock wall measurements (Supplementary Figures S1-S20).

### 4.2 Modelled ground temperature

### 4.2.1 Western Norway

***Mannen***: Most simulations indicate permafrost absence in Mannen since 1900 (Figure 4; Video 1). However, the coldest scenario ("T-1 °C") reveals that permafrost pockets could have existed in the mountain before 2019 in moderately steep slopes below the uppermost rock face, where the monthly SOs arising from solar radiation were not assumed (Video 1). The modelled GTs above 1100 m are mainly between 0.5 and 1.5 °C, with a maximum range of GTs in the uncertainty runs between 0.6 to 0.9 °C (Figure 4). The main ground heat flux direction is



almost vertically one-dimensional beneath the plateaus and tilts outwards towards the colder slopes elsewhere (Video 2).

*Hogrenningsnibba*: On the NNE-and SSW-facing slopes of Hogrenningsnibba, the lower permafrost limits remain respectively at about 1300 m and 1400–1450 m during the last 120 years (Figure 4; Video 3). The simulated GTs

are slightly lower at the lower permafrost limit on the SSW-facing slope due to the less-conductive, 1 m thick colluvium layer (compare "Main" with "Bedrock" in Video 3). GTs are above -3 ℃ with the range of GTs in the uncertainty scenarios below 1 ℃ (Figure 4). In most scenarios, the main heat flux direction is towards the colder NNE-facing slope, except for the simulation with the glacier (Video 4).

*Kvernhusfjellet*: According to the GT field simulated for Kvernhusfjellet (Figure 4), warm sporadic to

discontinuous permafrost occurs in the mountain, also below parts of the snow patch on the upper plateau and the glacier on the east-facing slope. In the west-facing slope, sporadic permafrost occurs down to an elevation of 1550 m in 1940, 1400 m in 1980, and degrades almost completely in 2020 with a deep permafrost limit at around 1620 m (Figure 4; Video 5). GTs are mostly above -1 ℃ with the modelled range of temperature in the uncertainty simulations of below 0.5 ℃ under the glaciated slope and 0.5–1 ℃ under the SW-facing slope (Figure 4). The

warm-based glacier contributes to slightly higher GTs in the mountain (Video 5). Since the simulated permafrost is warm (>-2 ℃), ignoring SOs results in a difference of several hundred metres in lower permafrost limit and changing the SAT forcing by adding 1 ℃ yields sporadic to no permafrost. In the "N logger" scenario, permafrost is modelled down to 1300 m in 2020, whereas in the warmer "S logger" scenario permafrost limit is at 1600 m in 2020. The main heat flux direction in the mountain is towards the coldest zones below the west-facing rock face

(Video 6). In the warmest scenarios, the heat flow direction in the middle of the mountain is, however, almost one-dimensional vertically towards the plateau. Even though Kvernhusfjellet and Hogrenningsnibba lie close together, permafrost limits are at a higher elevation at the W-E Kvernhusfjellet profile than at the SSW-NNE Hogrenningsnibba profile. This difference results from the extent of the steepest parts, where we applied SOs, and is particularly clear when comparing the "Main" simulations with the "Without monthly offsets" simulations

(Videos 3, 5), i.e. ignoring SOs at steeper Kvernhusfjellet leads to much lower GTs in the whole mountain than when ignoring SOs at moderately steep Hogrenningsnibba. Moreover, the similar scenarios with "Bedrock & Glacier at NNE" for Hogrenningsnibba and "S logger" for Kvernhusfjellet show how the differences in geometry influence permafrost distribution, e.g., permafrost limit is modelled at 150 m lower elevation in the former scenario.

*Ramnanosi*: The main run for Ramnanosi using the west-facing RW logger suggests no permafrost in the mountain since 1900 (Figure 4; Video 7). For 2020, GTs are mostly below 2 ℃ at elevations above 1200 m with the GT range below 1.1 ℃ (Figure 4). The three coldest sensitivity scenarios, namely "Without monthly offsets", "T-1 ℃" and "N logger" indicate that warm permafrost (>-2 ℃) has been present in the rock wall above an elevation of 1200 m over the last 120 years (Video 7). Modelled permafrost is in a degrading state in 2020 and its temperature

is above -0.5 ℃ in the coldest scenario. The main flux direction in Ramnanosi is usually towards the coldest zones somewhere in the upper parts between the rock wall and plateau in most scenarios (Video 8). For the colder scenarios, where rock walls have cold zones, heat flux is mainly directed towards them. In the "S logger" scenario with a much warmer rock face, heat flux is forced towards the colder zones parallel to the plateau surface, suggesting that plateaus may be colder than rock walls if SOs are large enough.



### 4.2.2 Jotunheimen

Figure 5 provides modelled maximum GT in the Jotunheimen profiles. For both profiles, sporadic to discontinuous permafrost is simulated down to an elevation of 1420–1520 m in 1980 and 1530–1590 m in 2020. Considering the simplified forcing for the gentle terrain in our modelling, a boundary between discontinuous and continuous permafrost can only be established assuming a particular isotherm, here -2 °C, as the lower limit for continuous permafrost. In that case, continuous permafrost starts at ~1780–1860 m in 2020 for the gentle terrain. For the highest plateaus (>2100 m), which are covered with blockfields, GT is modelled between -6– -4 °C. The simulated span of the GT in the Jotunheimen with respect to the main run is similar to the runs for the Western Norway, with the deep GTs up to 1.0 °C different from the main run (Figure 5). The bedrock beneath glaciers have the least GT span since we considered the same temperatures at glacier beds in all sensitivity scenarios. Sensitivity simulations display highest GTs for "Blockfields nF = 0.4", where snow conditions are changed substantially for the widespread blockfield-covered plateaus in the Jotunheimen (Videos 9, 10, 12, 13).

***Veslpiggen***: The assumed warm-based glaciers increase GTs at Veslpiggen and the ground below the thickest parts of the glaciers is modelled without permafrost. The ground below the thinner glacier parts is, nevertheless, underlain by permafrost. The assumed GST conditions at the blockfield locations have a large thermal influence on the deeper GT in the rock walls. In the NW-E Veslpiggen profile, the warmest scenario "Blockfields nF=0.4" indicates the coldest permafrost areas below the 150 m high NW-facing rock wall, whereas in the main scenario coldest permafrost is modelled below the blockfield-covered plateau (Video 9). Consequently, the main heat flux in the area has direction towards the coldest midsection below the rock wall in the scenario "Blockfields nF=0.4" (Video 11). Otherwise, the main heat flow is towards the coldest zone parallel to the surface topography of the plateau in the main scenario (Video 11). For the east-facing rock walls at Veslpiggen, GT in the rock walls frequently exceeds GT in the blockfields at a similar elevation due to the large SOs in this exposition, forcing the heat flux from the warmer rock walls towards the colder plateaus. Large SOs even in the NW-facing exposition and warm-based glaciers seem also to modify the GT in the Veslpiggen Plateau. The removal of glaciers leads to major changes in the main heat flux direction below the plateau from the tilted heat flux direction between the glaciated E-facing slope towards the colder blockfield-covered plateau in the main scenario to primarily vertically one-dimensional heat flux direction deeply in the mountain in the scenario "Without glaciers".

***Galdhøe***: The modelled GT in the Galdhøe Plateau is much less thermally affected by glaciers than the Veslpiggen Plateau and is almost the same in the main scenario and the scenario without glaciers (Video 12). The assumed forcing for the blockfields influences GT in the rock walls similar to the Veslpiggen Plateau. In the scenarios "Blockfields nF=0.4" and "Without monthly offsets" the main heat flux is directed towards the colder zones below the rock walls and even the east-facing rock wall is underlain by a relatively lower GT than the surrounding ground (Video 14). Most other scenarios mainly show a slightly tilted one-dimensional heat flow direction towards coldest permafrost below the blockfield-covered plateau. GT below blockfields is modelled colder than below till, which can be seen at one section along the Galdhøe profile, where we applied the till stratigraphy and nF-factor of 0.4 (Compare with Figure 2).

### 4.2.3 Northern Norway

***Gámanjunni***: Modelled maximum GT for Gámanjunni shows colder northeast-facing slope compared with the southwest-facing slope, and the lower permafrost limits are approximately 100 m higher at the southwest-facing





slope, at an elevation of around 850 m in 2020 (Figure 6). We note, however, that this asymmetry is not related to
the higher SOs applied to the SW-facing rock wall, and is rather caused by the extent of the steeper terrain in the
profile. The NE-facing slope is rougher and consists of several smaller rock walls, whereas the SW-facing slope
encompasses mainly one smoother rock wall, less than 50 m in height. The influence of geometry is especially
clear in the "W logger" scenario (Video 15), where we applied slightly colder forcing to the SW-facing rock walls
and the results still show lower deeper GT in the NE-facing slope. The modelled permafrost temperatures in the
main simulation are between -2 and 0 °C, barring the upper parts of the NE-facing rock wall where permafrost
colder than in -2 °C was modelled in the colder years. The sensitivity test maximum range in the modelled
maximum GT is mostly <1 °C, except for e.g. the lowest part of the unstable slope at Gámanjunni in some years
(Figure 6). Furthermore, the simulations indicate that the uncertainty in the water content is less important than
uncertainties in the temperature forcing or snow conditions (Video 15). The main heat flux direction in the
mountain is often modelled towards the coldest zone below the NE-facing slope, whose depth is deepening over
the last few years (Video 16). In addition, the results show that in the simulations with SOs, the scree slope is often
colder than the sun-exposed, SW-facing rock wall. The scree slope is also less coupled to atmospheric conditions
due to snow cover and greater ice contents, hence permafrost degradation occurs slower than in the rock wall,
further amplifying the differences in GT between the sun-exposed rock face and scree slope during the warmer
periods. In the simulation "Without monthly offsets", the rock wall is always colder than the scree slope.

*Ádjit*: At Ádjit, the modelled permafrost limits are lower on the southwest-facing slope than on the northeast-
facing one, at around 650 m in 1980 and 700 m in 2020, roughly where the active rock glacier has its front (Figure
6). The south-facing rock wall is warmer than the north-east-facing slope, where in the colder periods, cold
permafrost (<-2 °C) was modelled in the upper parts (Video 17). The maximum range of GT is generally below 1
°C; however, some parts close to or inside the rock glacier have a slightly higher GT span of up to 1.2 °C (Figure
6). For this profile, the SW-facing rock wall is much steeper than the moderately steep NE-facing slope above
1000 m elevation, which is the reverse of Gámanjunni geometry. The simulation "Without monthly offsets" (Video
17) shows the SW-facing slope as colder than the NE-facing slope due to the extent of the rock walls. Furthermore,
tested uncertainties in the moisture content affect the results much less than the uncertainty in the GST forcing and
slightly less than the uncertainty in the snow conditions. In the main scenario, the main heat flux direction is
towards the colder zones below the NE-facing slope (Video 18). The colder zones move rapidly; hence, the tilt of
the heat flux direction from the SW-facing rock wall undergoes some changes and is horizontal in the middle of
the rock wall in 2020.

*Rombakstøtta*: Permafrost limits at Rombakstøtta are modelled slightly higher than at the other sites in the
Northern Norway (Figure 6), at approximately 900–950 m and 1000 m for the NNE- and SSW-facing slopes in
both 1980 and 2020. Only warm permafrost with temperatures above -2 °C is modelled in the main scenario, with
the span of GT of less than 1.1 °C (Figure 6). Permafrost temperatures are slightly higher in some parts of the
NNE-facing slope because we only applied monthly SOs due to solar radiation on slopes steeper than 60° and the
SSW-facing slope is mainly <60° steep. In the scenario "Without monthly offsets" GTs are much lower on the
NNE-facing rock wall than on the SSW-facing slope (Video 19). The main heat flux direction in the main scenario
is towards the cold zone somewhere in the upper sections in the middle of the mountain, suggesting that the forcing
between the steeper NNE-rock wall and the SSW-slope is somewhat similar (Video 20). This balance is disturbed
in the scenario "Without monthly offsets", where a colder zone is modelled below the NNE-facing rock wall.



### 4.3 Ground temperature trends in rock walls

**4.3.1 Western Norway**

GTs at 20 m depth increased less than 0.2 °C decade$^{-1}$ at Mannen site between the 1900s and the 1930s, changed less than 0.05 °C decade$^{-1}$ between the 1930s and the 1980s, and increased again between the 1980s and the 2010s with a rate of 0.1–0.2 °C decade$^{-1}$ (Figure 7). For Hogrenningsnibba slight GT increase is modelled between the 1900s and the 1930s (Figure 7). 20 m GT remained similar between the 1930s and the 1980s. Between the 1980s and the 2010s, 20 m GT increased around 0.1–0.2 °C decade$^{-1}$ and the warming rates were larger than over the 460 previous decades. Kvernhusfjellet has similar cooling and warming trends as Hogrenningsnibba; however, with slightly higher rates below the steepest parts (Figure 7). Figure 7 also shows that the steepest parts of the profile are most responsive to both cooling and warming. Unlike the other sites in the Western Norway, Ramnanosi had a decreasing trend in SAT10a at the beginning of the 20th century (Figure B1); hence, also GT decreased slightly between the 1900s and the 1930s (Figure 7). 20 m GT at Ramnanosi had only small differences between the 1930s 465 and the 1980s and increased by 0.05–0.25 °C decade$^{-1}$ since the 1980s (Figure 7).

**4.3.2 Jotunheimen**

At the highest elevations in the Jotunheimen, GT at 20 m increased between the 1900s and the 1930s by less than 0.1 °C decade$^{-1}$, then remained similar between the 1930s and the 1980s, and raised by up to 0.35 °C decade$^{-1}$ 470 between the 1980s and the 2010s (Figure 8). Steep slopes are the most responsive to warming; however, GT in the blockfields on the highest plateaus is also strongly coupled with SAT in our simulations, since we applied a high nF-factor (0.7).

**4.3.3 Northern Norway**

Gámanjunni and Ádjit had the largest SAT10s rise at the beginning of the 20th century since 1900 (Figure B1), 475 therefore GT increase is larger between the 1900s and the 1930s than between the 1980s and the 2010s (Figure 9). Between the 1930s and the 1980s, GT slightly decreased at depths below 20 m and increased at depths deeper than 20 m in some areas due to rise in atmospheric temperature in the early 20th century. The rock walls are the most sensitive terrain type to GST trends. Ádjit has similar trends to Gámanjunni; however with larger rates, especially in the uppermost parts where 2D effects largely influence GT temperature (Figure 9). Rombakstøtta has similar 480 cooling and warming trends to the other sites in the Northern Norway; however, increases of both SAT10a and GT are larger since the 1980s (Figure 9).

### 4.4 Ground temperature at 20 m depth

**4.4.1 Elevation distribution**

Modelled GT at 20 m depth in the rock walls is often coldest in the scenarios "Without monthly offsets" or "T-1 485 °C" (Figure 10). Scenarios "Without monthly offsets" yield usually coldest midsection in a single rock wall, whereas most other scenarios differ from these results, except for the simulations using data from the north-facing loggers for Kvernhusfjellet and Ramnanosi, which have small average annual SOs (~0.5 °C). Highest GT at 20 m is most often modelled in the runs with "T+1 °C", hence SOs arising from solar radiation and SAT forcing are the





most important factors for the modelled GT within the tested values. In the following, we mention the remaining
thermal controls on modelled GT in the rock walls.

For the smaller NNE- or NE-facing rock walls at Hogrenningsnibba, Mannen, Ádjit, Gámanjunni, and
the lower (< 1550 m) smaller west-facing rock walls at Kvernhusfjellet, 20 m GT increases or decreases with
elevation depending on the distribution of the various terrain types in the vicinity of a single rock wall. GT
increases with elevation if terrain above a single rock wall is gentler than terrain below this single rock wall, and
the opposite is modelled if the terrain above is steeper than terrain below. Thus, 20 m GT distribution in such rock
walls is predominantly due to snow cover distribution in the rock wall vicinity. The larger NE-facing rock walls
at Rombakstøtta have in general decreasing GT with elevation; however, the mentioned small rock wall pattern is
superimposed on the latter pattern, e.g. the thermal influence of snow cover on the plateau, possibly with heat flux
from the SSW-facing rock wall, is evident in the topmost rock wall. The SSW-facing rock walls at
Hogrenningsnibba and Rombakstøtta also fit this pattern for small rock walls. For the west-facing Ramnanosi, GT
is highest in the lower part of the rock wall, because the assumed no SOs arising from solar radiation and little
snow on the plateau lead to lower GT in the upper part of the rock wall.

The SE-facing rock walls at Ádjit and Gámanjunni have somewhat different GT patterns. Ádjit has
pronounced higher GT in the middle of the rock wall and it seems that the SOs in the middle section dominate
other thermal influences. The small rock wall at Gámanjunni has a pattern of larger rock walls, i.e. the middle part
of the rock wall is either coldest ("Without monthly offsets") or warmest, indicating that the thermal influence of
snow is smaller.

The east-facing rock walls in the Jotunheimen have also various GT distributions at 20 m, depending on
the rock wall size. Larger rock walls (> 50 m high) have the highest GTs in their midsection in most scenarios,
except for the scenario "Without surface offsets", where the midsection is coldest, pointing to the large thermal
influence of SOs. The smaller, lower east-facing rock walls at Galdhøe have less steep terrain below than above
them, hence GT decreases with elevation. The small east-facing rock walls at an elevation of ~2200 m at
Veslpiggen are below blockfield-covered plateaus, therefore temperature decreases with elevation in most
scenarios, except where more snow was applied on the plateau. The uppermost east-facing rock wall at ~2300 m
at Veslpiggen has glaciers below and blockfields on the plateau, and GT decreases with elevation due to the large
thermal influence of the glaciers. The lower west-facing rock wall at Galdhøe is thermally influenced by the colder,
moderately steep terrain below and warmer gentler terrain above. The GT distribution in the upper west-facing
rock walls in the Jotunheimen is governed by the snow conditions on the blockfield-covered plateau, and GT
decreases in general with elevation in the scenarios with less snow on the plateau.

**4.4.2 Warming rates**

Over the last four decades, SAT at the rock wall elevations along the profiles increased by 0.25–0.4 °C decade[-1]
with the largest warming rates in the Jotunheimen and at Rombakstøtta (Figure 11 A, C, E, G). As mentioned
earlier, we reconstructed the same SAT10a trends along the profiles elevation-wise, so there is no elevation trend
in the results, although we cannot exclude such trends. We show the SAT10a pattern here only to compare it with
the modelled GT and we note that they might be inaccurate.

Trends of GT at 20 m depth have a more complex pattern elevation-wise (Figure 11 B, D, F, H); however,
the largest simulated values are still in the Jotunheimen and at Rombakstøtta. GT at 20 m depth increased on



average by 0.2 °C decade⁻¹. Ádjit has larger warming rates compared with Gámanjunni, especially at the higher
elevations, pointing to the increasing importance of the two-dimensionality since the former has a sharper peak.

The Jotunheimen has the largest mean 20 m GT increase (0.25 °C decade⁻¹), likely because we allowed the
blockfield-covered plateaus to be relatively strongly coupled with SAT, so the two-dimensional warming is more
effective in rock walls below plateaus. In general at all sites, within a single rock wall, warming rates seem to
increase towards the uppermost part of the rock wall. We simulated similar patterns in the previous model runs
using the seNorge data set when SAT increase rates sometimes decreased with elevation. It is expected that the

2D effects will increase with elevation in a single rock wall just based on the topography of the study sites. For a
2D profile, the distance from surfaces above a rock wall to a 20 m depth in a rock wall below is shorter than the
distance from surfaces below a rock wall to a 20 m depth in a rock wall above. Generally, ground warming rates
at 20 m depth seem to be independent of elevation (Figure 11 F) and slightly increase with latitude (Figure 11 H).
The latitude dependency is nevertheless the combination of the sharp peak Ádjit and the larger SAT increase at

Rombakstøtta, hence we note that there are only a few profiles included in this study and further studies are
required to investigate SAT, GST and GT trends in rock walls in Norway.

Furthermore, we show sensitivity of the modelled GT rise at 20 m depth in Figure 12. For most rock walls
warming rates increase with elevation as shown earlier. There are, nevertheless, a few exceptions:

(1) Warming rates decrease with elevation for rock walls that are slightly less steep in the upper parts.

(2) For parts of rock walls where permafrost thawed at 20 m depth between the 1980s and the 2010s, warming
rate is larger. Even small latent heat effects in permafrost slightly retard warming and this effect
disappears when permafrost is absent. For instance, the "N logger" scenario for Ramnanosi shows the
largest rates of warming in the lower rock wall, where GT is below 0 °C in the 1980s and above 0 °C in
the 2010s. For the upper rock wall, permafrost is still modelled in some years in the 2010s. The "nF-0.10"

and "T+1°C" scenarios for Ádjit, together with the "T+1°C" scenario for Hogrenningsnibba and
Rombakstøtta similarly show higher warming rates in portion of the rock walls where permafrost
degraded between the compared decades.

Glaciers reduce ground warming in nearby steep rock faces, e.g. the east-facing rock wall in the Jotunheimen has
higher GT increase in the scenario "Without glaciers". Otherwise, the assumed snow conditions have largest

influence on the warming rates, i.e. any snow accumulation in rock walls lead to lower warming rates. Snow cover
in the rock wall vicinity also influences the modelled warming rates, e.g. rock walls below plateaus or rock ledges
in the Jotunheimen have smaller warming rates if more snow is applied above them.

**5 Discussion**

**5.1 Limitations and strengths**

**5.1.1 Subsurface heat transfer**

The CryoGrid 2D model is based entirely on thermal conduction that is believed to be the dominant heat transfer
process in the ground (Williams and Smith, 1989). However, non-conductive thermal processes along
discontinuities and within the cracks, such as air convection or advection by moving water, might contribute to the
subsurface thermal regime (e.g. Draebing et al., 2014). Many discontinuities may exist in the bedrock and be

further widened by frost weathering processes, allowing for generating pathways for the advective heat transfer to





occur. The exact configuration of bedrock discontinuities allowing for including them in our modelling is unavailable. A study by Hasler et al. (2011a) in the Swiss Alps showed that while heat advection by percolating water has a negligible thermal impact, air ventilation likely causes thermal offsets similar to the offsets in coarse sediments and values of up to 3 °C are reported. Another study by Moore et al. (2011) analysed measured deep
GT profiles and attributed their disturbed profiles to localised convection cells in the fractures, whereas seasonal water infiltration had a minor influence on GTs. Nevertheless, several studies still emphasise the importance of advective heat input for GTs in permafrost-underlain terrain (e.g. Krautblatter and Hauck, 2007; Hasler et al., 2011b). A study by Magnin et al. (2017a) showed, however, that non-conductive thermal processes are only relevant in the upper 6 m below the ground surface. It is also noteworthy that conductive heat transfer in
discontinuities filled with ice would alter GTs, i.e. ice infills in permafrost could act as significant heat sinks; however, we cannot find any study that supports this hypothesis except for an open question in Gruber (2005).

Air convection is likely responsible for the observed negative thermal anomalies in the coarse-sediment landforms, such as blockfields (Heggem et al., 2005), rock glaciers (Wicky and Hauck, 2020) and talus slopes (Lambiel and Pieracci, 2008; Wicky and Hauck, 2017). Studies by Juliussen and Humlum (2008) and Gruber and
Hoelze (2008) show examples of how the conductive heat transfer could account for the negative thermal anomalies in the blockfields. Even though views of these authors on the governing mechanisms could be implemented in our model, responsible thermal processes are yet to be proven. In our study, negative thermal anomalies in the blockfields and rock glaciers are at least partly accounted for through the larger nF-factors than in the other sediment cover types.

Furthermore, the CryoGrid 2D model considers the 2D heat diffusion, which is an advance compared with the 1D case; nevertheless, heat transfer processes in complex terrain may occur three-dimensionally (Noetzli et al., 2007; Noetzli and Gruber, 2009). Myhra et al. (2017) argued that even this is the limitation of the CryoGrid 2D model, applying it to the Norwegian mountains with flat plateaus and long valleys could be adequate. Magnin et al. (2017a) employed a similar 2D model to ours and validated the data against rock wall boreholes. The authors
claimed that the 3D effects were likely of little importance for GT and the 2D modelling approach was sufficient for sharp topography in the European Alps. Nevertheless, our 2D approach could potentially underestimate the GT trends in areas where GST signal penetrates from more than two sides, as modelled in Noetzli and Gruber (2009).

### 5.1.2 Model forcing

The CryoGrid 2D model was forced using lapse-rate adjusted SATs, together with the measured average monthly
SOs in steep rock faces. The number of meteorological stations is low in the mountains in Norway; nevertheless, they still are well correlated with the rock wall logger data after adjustments for the monthly SOs. There are some uncertainties in lapse rates and the reconstructed long time forcing is especially uncertain. Moreover, we had to use the SeNorge data set for some sites, which is based on the spatial interpolation between the in situ data (Lussana et al., 2018).

Furthermore, we only force the model directly with GST, instead of including a surface energy balance, as for instance in Noetzli et al. (2007). We applied the same SOs to each year, based on the average offsets between GST and SAT, which could otherwise be modelled using surface energy balance. However, we lack data to be able to implement such an approach at the time scales used in this study. Snow cover and solar radiation are



believed to be the main controlling factors for GST in the rock walls (Haberkorn et al., 2015) and snow cover
governs the distribution of GST in the gentle terrain in Norway (Farbrot et al., 2011; Gisnås et al., 2014), hence
our methods account for the most important SOs measured in Norway. Magnin et al. (2017a) showed that a similar
approach, i.e. without energy balance and without consideration of snow accumulation in rock walls, was
appropriate to reproduce rock wall temperature at depths > 6 or > 8 m by comparing the modelled temperature
with the measured temperature profiles in boreholes. For shallower depths, additional effects of non-conductive
heat transfer and local snow accumulations that were ignored in the modelling caused substantial temperature
differences.

Our analysis of the 2 h temperature suggests that solar radiation is very probably the main controlling
factor for SOs in the Norwegian rock walls, as also shown in Magnin et al. (2019). Large increases in maximum
daily temperature can be seen in the rock wall temperature series, pointing to solar radiation as the dominant source
of energy that modifies GSTs. The north-facing slopes in Norway can receive enough shortwave radiation to have
mean annual SOs of around 0.5–1.5 °C (Figure 3), hence ignoring SOs would lead to much lower GTs even for
this exposition. Similar ranges of average SOs were measured in the small cliffs in the north-facing loggers in the
Northern Norway (Frauenfelder et al., 2018). Furthermore, we note that we did not apply non-nival SOs to
moderately steep slopes (< 60° gradient), since we doubt that the observed non-nival SOs are as large as in the
monitored slopes. For instance, Hasler et al. (2011a) suggested that late-lying snow reduces GST in moderately
steep slopes, due to a reduction in the incoming shortwave radiation.

### 5.1.3 Snow distribution

One of the CryoGrid 2D model limitations is the lack of a snow domain; hence, we apply nF-factors for the gentle
and medium-steep terrain. Preferably, snow depth should be rather described dynamically both temporally and
spatially, including snow redistribution by avalanching and wind. However, research concerning snow distribution
is steep rock walls in Norway is lacking, so there are large uncertainties in snow depth and its timing. The studies
that we reviewed from elsewhere had some contrasting results about snow distribution in the steep rock walls: 1)
some studies point to that steep slopes above a certain threshold (e.g. more than 45º, 50º, 60º or 70º) cannot
accumulate permanent snow cover due to avalanching or wind drift (Blöschl et al., 1991; Kirnbauer et al., 1991;
Blöschl and Kirnbauer, 1992; Winstral et al., 2002; Machguth et al., 2006), 2) other studies, often using airborne
or terrestrial laser scanning, show that almost any slope gradient can accumulate snow (Wirz et al., 2011; Sommer
et al., 2015). The latter group of studies, nevertheless, recognises that snow cover is limited in steeper terrain and
accumulates less snow than gentler terrain. Furthermore, the studies use various parameters as the most crucial to
explain snow distribution in steep terrain, e.g. : 1) summer slope angle (Blöschl and Kirnbauer, 1992; Sommer et
al., 2015), 2)  terrain-wind-interaction (Winstral et al., 2002; Wirz et al., 2011), 3) elevation and terrain roughness,
which possibly correlates with the summer slope angle (Lehning et al., 2011). We note, however, that we used a
high resolution DEM of at least 1 m resolution to construct each profile, and 1 m DEM was considered precise
enough to detect rock ledges in the Swiss Alps, where snow can accumulate (Haberkorn et al., 2017), and such
areas have snow cover in our study. Snow distribution in rock walls in Norway remains to be quantified, e.g. using
LIDAR-scanning, and its governing factors recognised.



### 5.1.4 Thermal influence of snow

Snow cover could either warm or cool the ground. The overall effect of snow cover on GT is complex because it depends on e.g. snow thickness, depth, duration, timing, melting processes within a snowpack, snow structure (Zhang, 2005) or sun exposure (Magnin et al., 2017b). Snow cover affects GT in both steep and gentle terrain in multiple ways:

(1) As an additional buffer layer with low thermal conductivity, snow insulates the ground, given that SAT is lower than GT and snow cover is sufficiently thick, e.g. at least 0.6 m in the gentle terrain (Luetschg et al., 2008) or even 0.2 m in the rock walls (Haberkorn et al., 2015). This is likely the most important net thermal impact of snow on the GTs in Norway. Observed differences between GST and SAT are positive at most permafrost sites in Norway (Farbrot et al., 2011) and as shown in this study (Figure 3) all measured mean annual SOs in the rock walls are positive, hence the overall cooling of the ground surface annually due to snow cover is not observed in Norway. We note that the installed rock wall loggers in Norway should measure only snow-free rock walls by design (Magnin et al., 2019), hence, the available measurements are insufficient to preclude cooling due to snow cover.

We assumed that rock walls are snow-free, because our analysis of the measured rock wall temperature in Norway indicates only minor thermal influence of snow, as also mentioned in Magnin et al. (2019). We note, however, that the computed mean monthly SOs (Figure 3) account also for thermal effects of snow cover if there are any, hence rock walls are not sensu stricto snow-free in this study. For instance, W- and N-facing loggers at Gámanjunni are approximately 1 °C higher than the south-facing logger (Figure 3) in December and January, which is likely due to snow cover. The rock wall loggers at Rombakstøtta are probably the most influenced by snow from all loggers in Norway, e.g. W-facing logger is colder than the N-facing logger in May (Figure 3), and both E- and W-facing loggers show sometimes much smaller standard deviation of daily temperatures compared with the N-facing logger, which is likely the least snow-influenced logger in this area.

(2) Snow cover increases albedo of the surface and thus reduces absorbed short-wave radiation, i.e. late-lying snow will cool the rock wall in the sunny conditions because sun rays cannot reach it (e.g. Hasler et al., 2011a; Magnin et al., 2017b). This cooling effect was concluded to be a major cooling mechanism on the thinly snow-covered rock walls in the Mont Blanc Massif (Magnin et al., 2015). However, this cooling hypothesis was concluded to be of little importance in the study by Haberkorn et al. (2017), who show that sunny snow-covered rock walls are always warmer than snow-free rock walls due to reduced ground heat loss in winter, i.e. point (1) above.

(3) High emissivity of snow increases the outgoing longwave radiation; however, its high absorptivity has the opposite effect, hence their impact on snow temperature is influenced by atmospheric conditions (Zhang, 2005).

(4) Snow requires large energy inputs to melt, hence GT will be lower than SAT during snowmelt; however, this usually lasts for a short time and might be unimportant on yearly time scales (Zhang, 2005). However, meltwater percolating inside the cracks can refreeze and act as an additional heat source or favour an accelerated melting of the cleft ice (Hasler et al., 2011b).

(5) During autumn, a thin snow cover could lead to an enhanced conductive heat flux from the ground due to the large thermal gradients between the cooled snow surface and warmer upper ground layers (Keller and Gubler, 1993; Luetschg et al., 2008).

(6) Deposition of snow might reduce ventilation effects in clefts (Hasler et al., 2011a).



(7) If snow accumulates under the rock wall or in the rock ledges, the incoming short-wave radiation may be
reflected diffusively towards snow-free parts of the rock wall, hence warming it. The latter effect is less
investigated in the permafrost studies, although its importance was emphasised in the surface energy balance
modelling of the high-arctic rock walls in Svalbard in Schmidt et al. (2021) and mentioned in Fiddes et al. (2015).
We speculate that the reflected shortwave from the surrounding snow-covered surfaces may be important in some
rock wall aspects in Norway. Measured temperatures in rock walls during winter show temperature increase, seen
in the 2 h temperature measurements as a distinct distribution due to the shortwave solar radiation, even in February
in the Northern Norway that we consider is connected with snow accumulation in the surrounding terrain. A similar
temperature increase is not observed at the same magnitude during autumn when snow is present less often. We
recognise, however, that this seasonality could be related to cloud cover, issues with lapse rate or cooling effects
of thin snow cover during autumn. Additionally, rock walls just above glaciers, e.g. in the Jotunheimen, might be
likely affected by reflected solar radiation from the glaciers all year round, and measurements from the east-facing
rock walls just above glaciers show particularly large SOs (Figure 3). Hasler et al. (2011a) also mention that the
south- and east-facing rock faces above glaciers in the Swiss Alps experience extreme solar radiation.
Nevertheless, the observed SOs in the Jotunheimen could be a result of the dark surface of the rocks in this area,
which have a lower albedo compared with the bedrock at the other sites presented in this study.

**5.2 Comparison with other studies**

Since we used the rock wall loggers as data for calibration of the forcing input and they only represent near-surface
temperatures, we only compare them with our modelling results qualitatively in this section, i.e. by assuming that
the mean temperature in a rock wall logger of 0 °C indicates permafrost. We additionally compare our results with
the statistical modelling presented in Magnin et al. (2019) for the period 1981–2010 (Figure C1), and these results
agree quite well with the modelled GT. We note, however, that this reference data should not be thought of as
validation data for deeper GTs and merely represent surface conditions, because: (1) Neither the rock wall
temperatures nor the statistical modelling account for the temperature offsets deeper in the ground, e.g.
measurements conducted by Hasler et al. (2011a) in the European Alps were even 3 °C lower at depth than mean
annual rock surface temperatures, hence the existing surface information might be insufficient, (2) as discussed in
Noetzli et al. (2007) and Noetzli and Gruber (2009), permafrost may occur below the slopes where surface
information indicates permafrost-free ground, because of lateral heat fluxes and/or the preservation of long-term
temperature signals at greater depths.

**5.2.1 Western Norway**

Mean temperatures measured in the rock wall loggers at Mannen are 1.27 °C (Aug 2015-Jul 2018; 1290 m) and
2.55 °C (Aug 2015–Jul 2020; 1290 m) for the north- and east-facing loggers, respectively. Mean measured rock
wall temperature in the west-facing logger at Ramnanosi is 1.55 °C (Aug 2016-Jul 2020; 1370 m). Hence, all
loggers at Mannen and the W-facing logger at Ramnanosi suggest an unlikeliness of permafrost presence in these
rock wall expositions over the last few years. The north-facing logger at Ramnanosi measured the mean rock wall
temperature of 0.02 °C (Aug 2016–Jul 2020; 1370 m); hence, permafrost was likely in the north-facing parts of
the slope at least before the measurement period started. Since some cracks exist on the plateau above Mannen
(Saintot et al., 2012) and Ramnanosi, air ventilation could lower GT in the area; however, since thick snow cover



accumulates on the Mannen Plateau, plugging of the cracks with snow could prevent air ventilation. The modelled GT for Mannen differs slightly from the results shown in Etzelmüller et al. (2021), where the seNorge data were used as forcing and SOs were ignored.

Measured mean temperature in the rock wall loggers in the Loen area was -1.77 °C (1709 m), 1.40 °C (1648 m) and 1.76 °C (1662 m) for the period Aug 2015–Aug 2018 for the north-, south- and west-facing loggers, respectively. The north-facing logger indicates that permafrost is likely, hence it agrees well with the modelled GT in this exposition. The west- and south-facing loggers have positive temperatures; however, it does not preclude that permafrost cannot exist in the mountain, because the modelled GTs are higher in the uppermost parts than in the middle parts along the rock wall, due to the influence of the thick snow on the plateaus. Furthermore,
modelled permafrost in Kvernhusfjellet is clearly degrading over the last few years and possibly the same is the case for the SSW-facing slope at Hogrenningsnibba.

### 5.2.2 Jotunheimen

In the Jotunheimen mean measured RW temperature is -1.77 °C (Sep 2010–Jul 2020; 2320 m), -2.15 °C (Sep
2010–Jul 2020; 2204 m), -2.23 °C (Sep 2010–Jul 2020; 2226 m), -3.55 °C (Sep 2010–Sep 2018; 2179 m) for the east-facing higher ("Eh"), east-facing lower ("El"), south-facing, and west-facing loggers, respectively. Most loggers indicate that even cold permafrost exists in the Jotunheimen Mountains, hence they agree quite well with the modelling results. We also compared the modelled GTs and deeper warming rates with the available borehole data and the results agree quite well (not shown in this study), although there are variations in snow conditions
between the boreholes, hence we compared the measurements with the various sensitivity scenarios.

### 5.2.3 Northern Norway

In the Gámanjunni area, RW loggers measured -0.08 °C (1220 m), -1.31 °C (1243 m), -1.62 °C (1183 m) in the south-, north- and west-facing loggers in the period Aug 2015–Sep 2020. Hence, at least warm permafrost conditions can be expected in the uppermost parts along the profile, as our model reproduces. The results shown
in Etzelmüller et al. (2021) for Gámanjunni show somewhat different subsurface GT field, because the seNorge data were used there as forcing and SOs were unaccounted for.

For Ádjit measured RW temperatures are -0.01 °C (1245 m) and -1.80 °C (1230 m) for the south- and north-facing loggers in the period Aug 2015–Sep 2020. Both loggers indicate permafrost, although the south-facing rock wall is close to non-permafrost conditions. However, permafrost is still possible deep in the mountain,
as modelled in our simulations, where the 2D effects modify the subsurface thermal field. Three-dimensional GT modelling of Polvartinden Mountain, around 30 km northeast of Ádjit, suggested lower permafrost limits at 600–650 m over the last few years (Frauenfelder et al., 2018), which agrees well with our results.

Rombakstøtta loggers have mean RW temperature of 0.10 °C (1228 m), -0.71 °C (1224 m), -0.96 °C (1208 m) for the east-, north- and west-facing loggers. All loggers for the east-facing logger at Rombakstøtta
indicate that at least warm permafrost might be present in the rock walls, which agrees well with the modelled GTs.





### 5.3 Thermal regime in steep slopes

Due to the strong coupling of GST and SAT in rock walls, rock walls might have lower GT compared with the surrounding terrain and permafrost aggradation might occur in them much faster than in the other types of terrain in the decreasing SAT conditions, as e.g. shown in the previous modelling study by Myhra et al. (2017). However, sun-exposed large rock walls might allow more heat to enter the mountain. One example is Kvernhusfjellet, where the lower limit of permafrost is at 1620 m over the last few years, which is higher than at the moderately steep Hogrenningsnibba, where permafrost limit is at 1450 m over the last few years. In Norway, permafrost research on moderately steep terrain is yet to be conducted, since there are large uncertainties in both snow distribution and SOs in moderately steep terrain in Norway. However, our results are in agreement with the conclusions in Magnin et al. (2019) that the permafrost limits might be higher in the sun-exposed rock walls than in the less steep terrain.

We constructed meshes for various topographies and extended the previously presented 2D modelling for Norway (Myhra et al., 2017), mainly by including SOs. While the previous results mostly showed the midsection along a single rock wall as the coldest, our simulations show the midsection, or more precisely the lower portions of the midsection, sometimes as the warmest along the rock wall (at 20 m depth), barring the north-facing rock walls. The sensitivity scenarios where we skipped SOs show the same results as in Myhra et al. (2017) with the much colder midsections. Because the rock wall data from Norway indicated average yearly SOs of at least 0.5 ℃, the colder midsections in the north-facing slopes are less pronounced in the main scenarios when compared with the scenarios without SOs. Our results also show that scree slopes might be warmer than rock walls if SOs are large enough, e.g. 3 ℃. The latter is in discordance with the study by Myhra et al. (2019), where rock walls had a cooling effect on scree slopes; however, we note that they still agree for rock walls with minimal SOs. The simulated subsurface thermal fields are more similar to the 3D modelling from the European Alps (Noetzli and Gruber, 2009). Hogrenningsnibba has the most similar geometry to the one presented in the study from the European Alps. Our simulations show quite similar distribution of the isotherms to the ones from the European Alps, except that the isotherms inside Hogrenningsnibba are less inclined. This is expected since the rock surface temperature difference between the north- and south-facing slopes is smaller than in the European Alps, as discussed in Magnin et al. (2019). Slope steepness is, however, also an important factor influencing the subsurface thermal field. Ádjit is the steepest slope presented in this study and although the measured mean annual GST difference between the north- and south-facing slopes is below 2 ℃, almost horizontal heat flux direction between them is often modelled. This suggests an increasing sensitivity of the subsurface thermal fields to small differences in forcing for the steep terrain. For instance, the modelled subsurface thermal field for the nearby less-steep Polvartinden indicates almost horizontal isotherms (Frauenfelder et al., 2018). We note, however, that the differences in SOs for various aspects presented in the latter study are smaller, around 1 ℃.

The importance of multi-dimensionality for the rates of GT rise was previously investigated in the studies by Noetzli et al. (2007) and Noetzli and Gruber (2009), where it was shown that surface warming penetrates steeper topography from several sides, thus leading to a faster pace of ground warming compared with flatter topography. Our study suggests also that multi-dimensionality is an important factor, although we only investigated 2D case. The modelled warming rate of on average 0.25 ℃ decade$^{-1}$ in rock walls in the Jotunheimen is slightly higher than the warming rate of 0.2 ℃ decade$^{-1}$ measured at 20 m depth in the deep borehole at Juvvasshøe since 1999 (Smith et al., 2021). GT in this borehole is highly coupled with SAT, and the borehole has nF-factor of around 0.9.



## 6 Conclusions

From this study, the following conclusions could be drawn:

(1) Discontinuous permafrost likely occurs along most modelled profiles, except for Mannen and Ramnanosi. Nevertheless, convective heat transfer along discontinuities at both Mannen and Ramnanosi could lower GT; hence, both sites could be underlain by sporadic permafrost. Rock walls at the highest elevations in the Jotunheimen are in the continuous permafrost zone.

(2) Rock walls in the Northern Norway experienced larger GT variations after LIA than rock walls in southern Norway, since both the 1930s atmospheric warming and the 1970s–80s cooling were more pronounced in the north. All simulations show increasing GT since the 1980s. Rock walls in Norway are warming at the rates of 0.2 °C decade$^{-1}$ on average at 20 m depth over the last three decades.

(3) Many of the modelled sites lie close to the lower boundary of mountain permafrost, hence the modelled GT is sensitive to the changes in the forcing. Within the tested forcing, uncertainties in the SAT leaded to the largest changes in the modelled GT. Neglecting SOs might lead to much lower GT in the rock walls, even in Norway.

(4) The rock wall exposition and its size seem important modifying factors for the permafrost distribution in the mountains. High rock walls, higher than 50 m, or several small rock walls (<50 m high) allow effective ground cooling and lead to lower permafrost limits in the mountain if SOs are not too large (e.g. Gámanjunni). High rock walls or several small rock walls might also allow more heat to enter a mountain and frequently sun-exposed rock walls may even have higher permafrost limits than moderately steep terrain (e.g. Kvernhusfjellet). GST forcing in smaller rock walls influences GT more locally, e.g. if they have large SOs, the thaw depth is deeper.

(5) The elevational distribution of GT at 20 m depth is influenced by the assumed snow conditions above and below rock walls. This effect is especially pronounced for smaller rock walls. Larger rock walls and sometimes even smaller rock walls might have coldest or warmest midsection depending on SOs. The north-facing rock walls have usually small SOs, hence their midsection is coldest. The rock walls with large SOs have warmest midsection.

(6) Multi-dimensional thermal effects inside the mountains are smaller in Norway than in the European Alps. This is the combined result of the (1) differences in mountain geometry, which in Norway are usually mountain peaks, arêtes of smaller relief than in the Alps or deep valleys, (2) GST differences in rock walls between the various expositions are not as pronounced as in the Alps. The steepest mountains in Norway are, however, sensitive to even small differences in GSTs between the various expositions.

(7) Ground heat flux is modified in rock walls in the Jotunheimen by blockfields and large glaciers. GST in blockfields may be relatively strongly coupled with SAT, leading to lower GT and higher rates of GT increase (at 20 m depth) in rock walls close to blockfields. Large glaciers decrease GT increase in the nearby parts of rock walls; however, in view of their potential future retreat, warming rates might increase in the closest parts of rock walls.

(8) In rock walls with large SOs, plateaus above or talus below might be colder than rock wall, forcing ground heat flux towards colder plateaus or talus slopes.



(9) North-facing rock walls could even be warmer than moderately steep south-facing rock walls. Nevertheless, this effect requires further studies to be confirmed.

**Appendices**

**Appendix A. Thermal conductivity.**

**Table A1. Thermal conductivity for the mineral fraction.**

| **Mountain, municipality** | Thermal conductivity [W m$^{-1}$ K$^{-1}$] |
| --- | --- |
| Mannen, Rauma | 2.5 |
| Hogrenningsnibba, Stryn | 2.3 |
| Kvernhusfjellet, Stryn | 2.3 |
| Ramnanosi, Aurland | 3.1 |
| Veslpiggen, Lom | 2.7 |
| Galdhøe, Lom | 2.7 |
| Gámanjunni 3, Kåfjord | 2.9 |
| Ádjit, Storfjord | 2.9 |
| Rombakstøtta, Narvik | 2.9 |



**Appendix B. Surface air temperature trends**

Atmospheric temperature has in general had an increasing trend in Norway since the end of the LIA. Figure B1 shows the 10 year running mean surface air temperature (SAT10a) evolution for the highest elevations along each profile. In the first decade of the 20th century, SAT10a were -0.59 to -1.75 °C lower than over the last 10 year period (2011–2020).

The warming during the early 20th century was largest in the Northern Norway, which experienced at 845 least 1 °C warming between the 1900s and the 1930s, whereas the Western Norway had around 0.4–0.7 °C warming in the same period. Ramnanosi is the site with the largest cooling trend at the beginning of the 20th century. The Jotunheimen had only small cooling between these decades. SAT10a was 0.5–0.7 °C lower in the Northern and Western Norway, respectively, between the 1930s and the 1980s. In the Jotunheimen, SAT10a increased between the 1930s and the 1980s by around 0.4 °C, although we note that there was a slight cooling in 850 the area in the early 1980s; however, it vanishes when the results are presented as a mean value for the whole 1980s. SAT10a increased by 0.86–1.16 °C at all study sites after the 1970s–1980s cooling. The recent warming is the largest in the Jotunheimen and at Rombakstøtta.

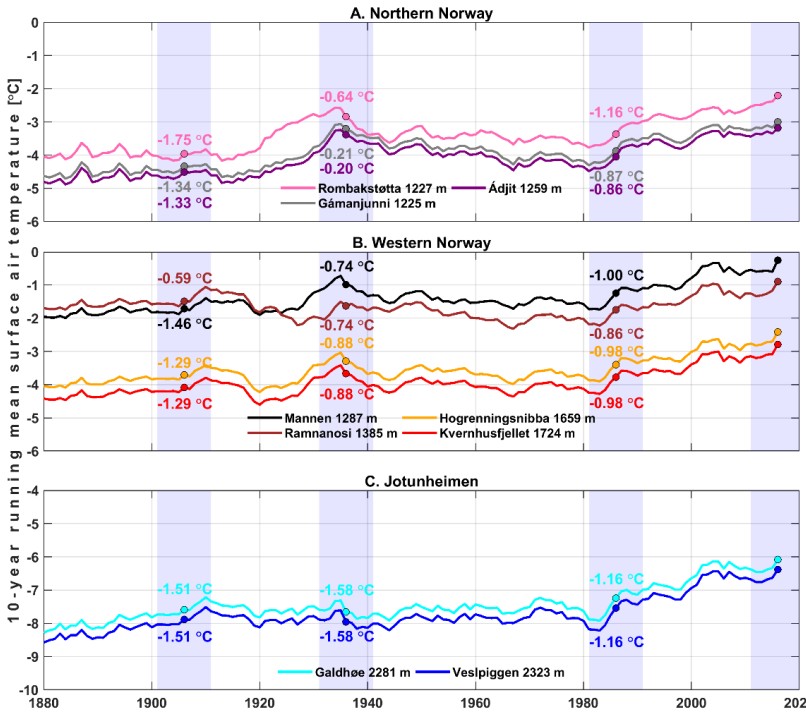

**Figure B1. 10-year running mean surface air temperature (SAT$_{10a}$) for peak elevations along each of the constructed profile in the Northern and Western Norway, together with Jotunheimen. Numbers along the plot lines are mean decadal temperature offsets in the 1900s, 1930s and 1980s relative to the 2010s.**





**Appendix C. Model comparison.**

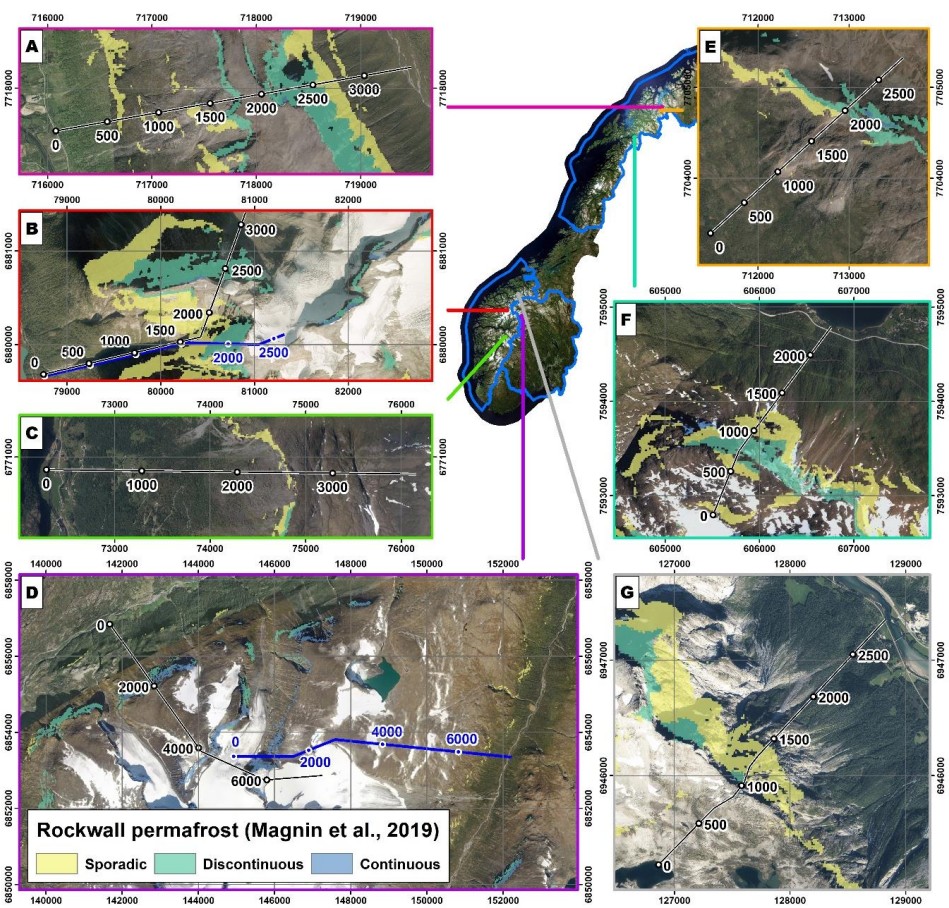

**Figure C1. Rock wall permafrost distribution according to Magnin et al. (2019) for: A) Gámanjunni 3, Kåfjord, B) Hogrenningsnibba (the northernmost profile/the black line) and Kvernhusfjellet (the southernmost profile/the blue line), Stryn, C) Ramnanosi, Aurland, D) Veslpiggen (the southernmost profile/the black line) and Galdhøe (the northernmost profile/the blue line), the Jotunheimen Mountains, E) Ádjit, Storfjord, F) Rombakstøtta, Narvik, and G) Mannen, Rauma. Numbers along the profiles indicate distance in metres. Map background credits: © Statens kartverk, Geovekst og kommunene. Coordinates in UTM zone 33N are shown.**

**Data availability**

Data are temporarily available through the University of Oslo's OneDrive cloud storage and will be later uploaded

to Zenodo: https://uio-my.sharepoint.com/:u:/g/personal/justync_uio_no/EULl5I7AKY1NtTmAw-SBSuwBWZh0bY2gCW_S6Wf4HVM-Yw?e=CiQ2gW (size ~38 GB).



**Author contribution**

JC performed the simulations and prepared the manuscript with contributions from all co-authors. BE prepared the first version of the forcing data and supervised the study. JC, BE and SW contributed to the conceptualisation of this study and developed the methods. KI prepared the regional SAT data sets and contributed to the analysis of the SAT trends. FM contributed to the discussion on the thermal regime in steep slopes.

**Competing interests**

The authors declare that they have no conflict of interest.

**Funding**

This study was funded through Justyna Czekirda's doctoral research fellow position at the Department of Geosciences, University of Oslo, Norway. Additional funding was provided by the project 'CryoWALL – Permafrost slopes in Norway' (243784/CLE) funded by the Research Council of Norway.

**Acknowledgments**

This study is based on rock wall loggers installed by Tobias Hipp and Bas Altena within CRYOLINK project in 2011, together with rock wall loggers installed later during the project 'CryoWALL – Permafrost slopes in Norway' (243784/CLE). Both projects were funded by the Research Council of Norway. Installation and data retrieval from the loggers were actively supported by the Geological Survey of Norway (NGU) and the Norwegian Water and Energy Directorate (NVE). Particular thanks are due to Reginald L. Hermanns (NGU) and Lars Harald Blikra (NVE). Paula Hilger, Thorben Dunse (both the University College of Western Norway), Ove Brynhildsvoll, Trond Eiken, Jaroslav Obu, Bas Altena, Juditha Aga, Harald Wathne Hestad and Erling Thokle Hovden (all University of Oslo) helped with retrieving rock wall logger data. Ole Einar Tveito from the Norwegian Meteorological Institute assisted in preparing climatic data. Kristin Sæterdal Myhra (the University College of Western Norway) provided the CryoGrid 2D code, while ice thickness data for the selected glaciers in Norway were provided by Liss Marie Andreassen (NVE). We want to thank the mentioned institutions and individuals.

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

**Figures**

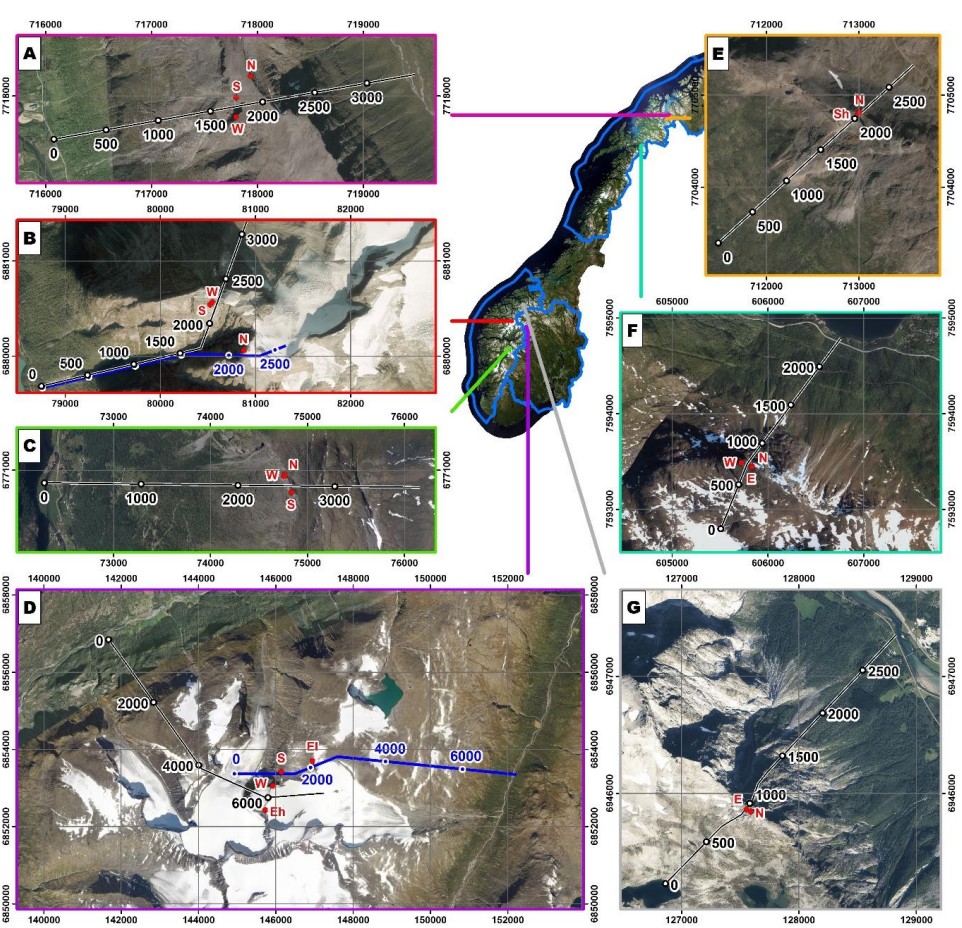


Figure 1. Transects for the two-dimensional modelling. A) Gámanjunni 3, Kåfjord, B) Hogrenningsnibba (the northernmost profile/the black line) and Kvernhusfjellet (the southernmost profile/the blue line), Stryn, C) Ramnanosi, Aurland, D) Veslpiggen (the southernmost profile/the black line) and Galdhøe (the northernmost profile/the blue line), the Jotunheimen Mountains, E) Ádjit, Storfjord, F) Rombakstøtta, Narvik, and G) Mannen, Rauma. Red points with
letters depict rock wall loggers in the various expositions: N=north-facing logger, S=south-facing logger, E=east-facing logger, W=west-facing logger, suffix "l"=at a lower elevation, suffix "h"=at a higher elevation. Numbers along the profiles indicate distance in metres. Three geographical regions of Western Norway, Eastern and Northern Norway are outlined by the blue lines. Map background credits: © Statens kartverk, Geovekst og kommunene. Coordinates in UTM zone 33N are shown.








**Figure 2. Slope geometry and stratigraphy: A) Galdhøe, B) Veslpiggen, C) Mannen, D) Gámanjunni 3, E) Kvernhusfjellet, F) Hogrenningsnibba, G) Rombakstøtta, H) Ramnanosi, I) Ádjit. The small case letters are stratigraphy codes described in detail in Table 2. The label "c/a" indicates alternating stratigraphy of bedrock and thin colluvium. Blue patches depict glaciers or perennial snow. Note that the meshes extend down to 6000 m below sea level and the parts below valley bottoms are not shown.**

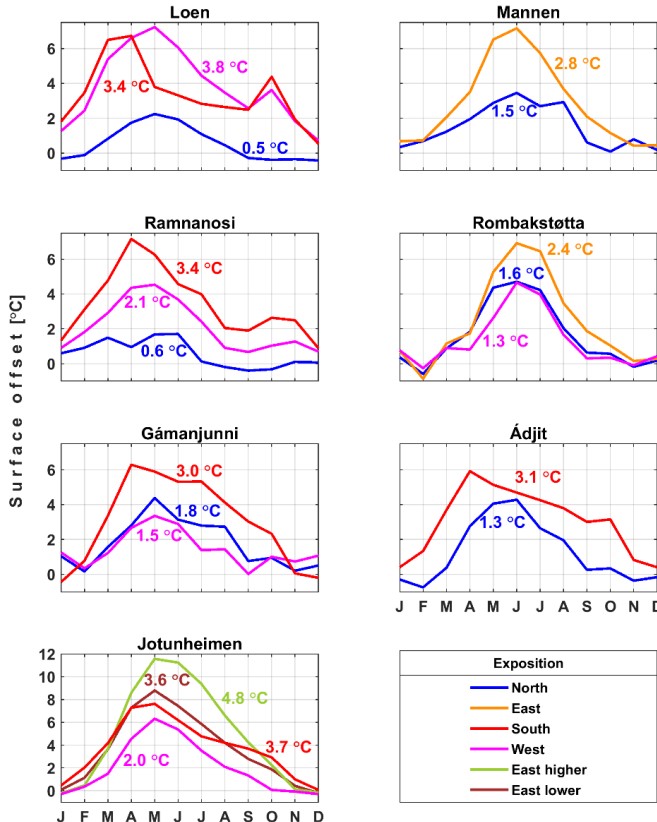

**Figure 3. Monthly surface offsets between air and rock wall temperature for each site and logger exposition. Numbers along the plot lines are average values. X-axis contains initials for months. Note that Jotunheimen has different y-axis than the other subplots. For Ádjit only the upper south-facing ("Sh" in Figure 1.) logger is shown.**




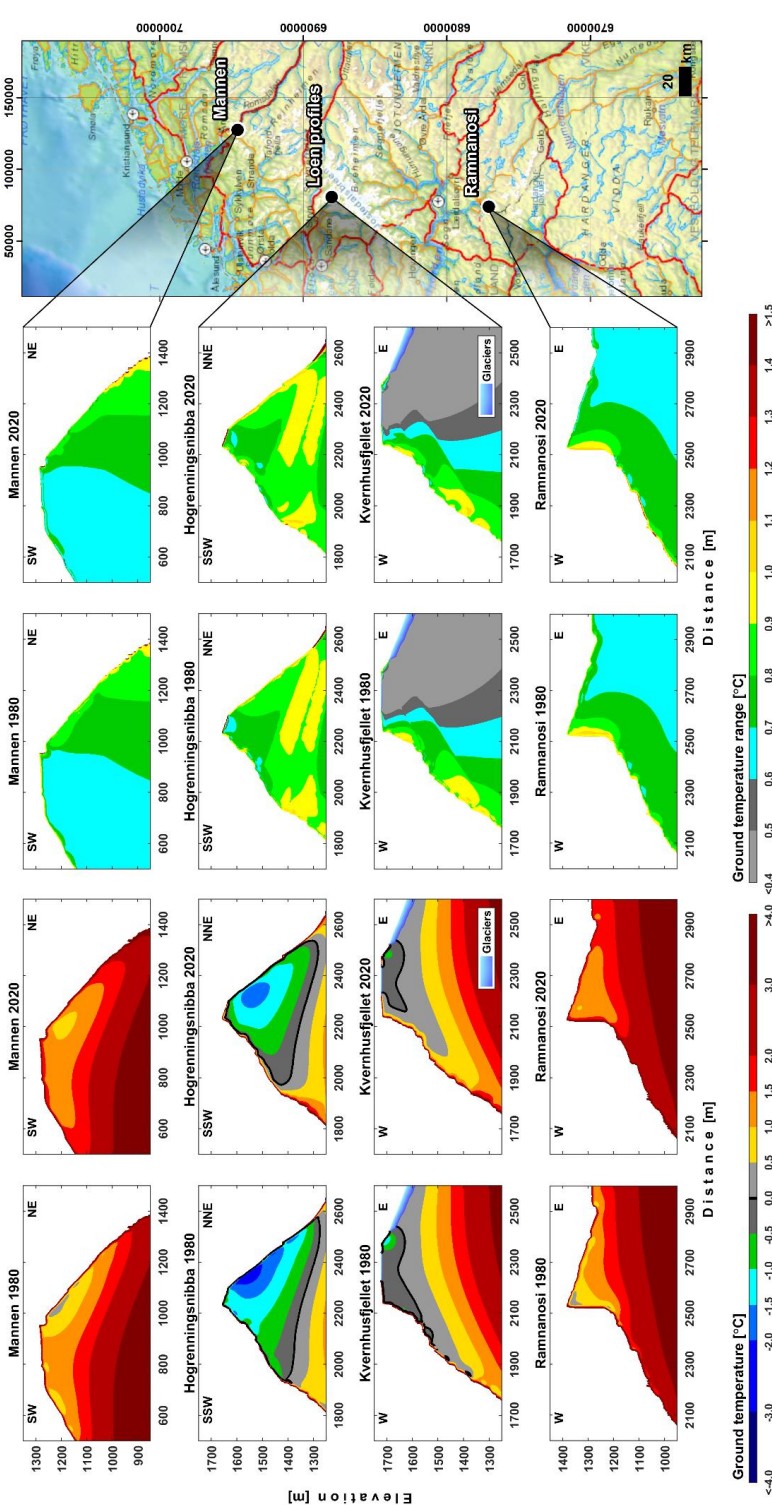

**Figure 4.** Modelled maximum GT and sensitivity test maximum range in the modelled maximum GT for the profiles in the Western Norway in 1980 and 2020. Maximum absolute differences in comparison to the main run are shown based on the uncertainty simulations. Map background credits: © Statens kartverk, Geovekst, kommuner og OSM – Geodata AS.








Figure 5. Modelled maximum GT and sensitivity test maximum range in the modelled maximum GT for the profiles in the Jotunheimen in 1980 and 2020. Maximum absolute differences in comparison to the main run are shown based on the uncertainty simulations. Map background credits: © Statens kartverk, Geovekst, kommuner og OSM – Geodata AS.



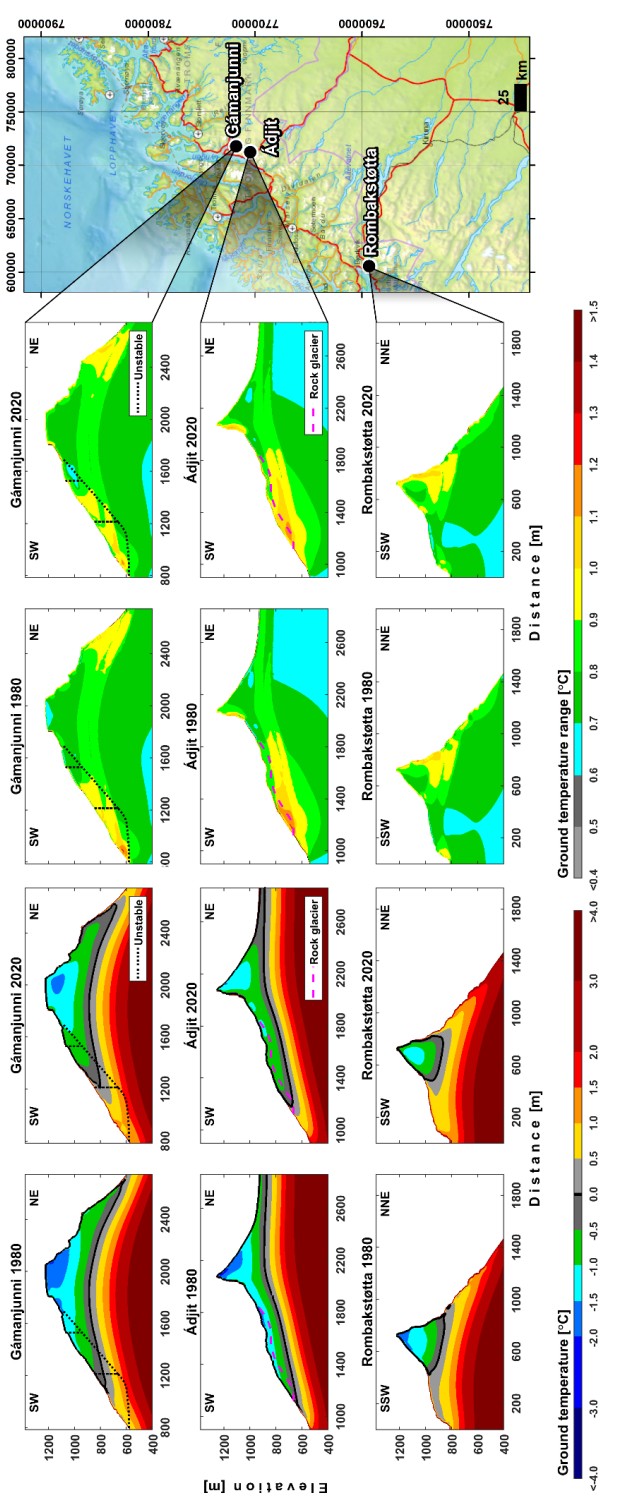

**Figure 6. Modelled maximum GT and sensitivity test maximum range in the modelled maximum GT for the profiles in the Northern Norway in 1980 and 2020. Maximum absolute differences in comparison to the main run are shown based on the uncertainty simulations. Map background credits: © Statens kartverk, Geovekst, kommuner og OSM – Geodata AS.**




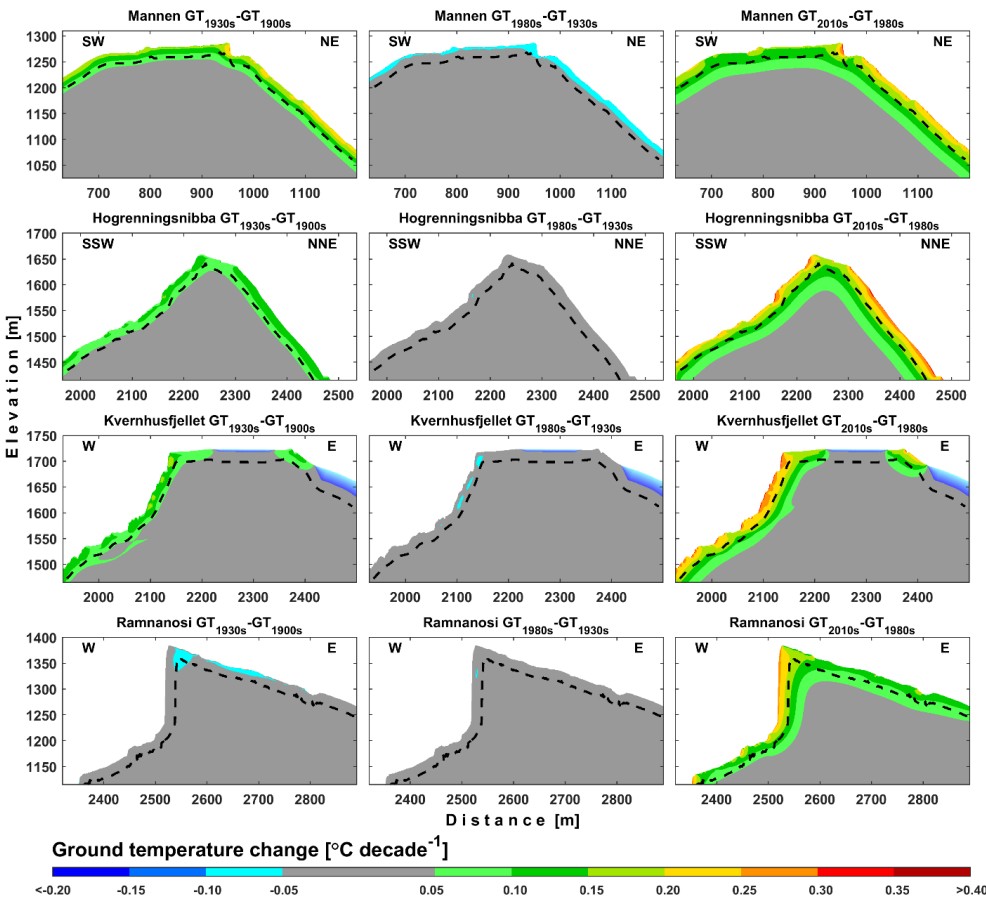

**Figure 7. Rate of change in 10 year mean GT for the profiles in the Western Norway between the following decades: (1) the 1900s and the 1930s, (2) the 1930s and the 1980s, (3) the 1980s and the 2010s. 20 m depth is delineated by the black dashed lines.**

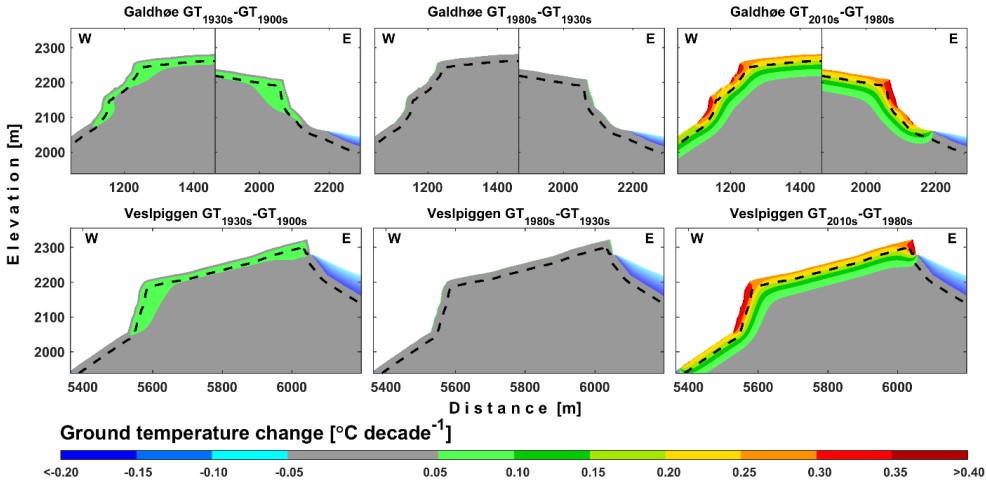


**Figure 8. Rate of change in 10 year mean GT for the profiles in the Jotunheimen between the following decades: (1) the 1900s and the 1930s, (2) the 1930s and the 1980s, (3) the 1980s and the 2010s. 20 m depth is delineated by the black dashed lines.**

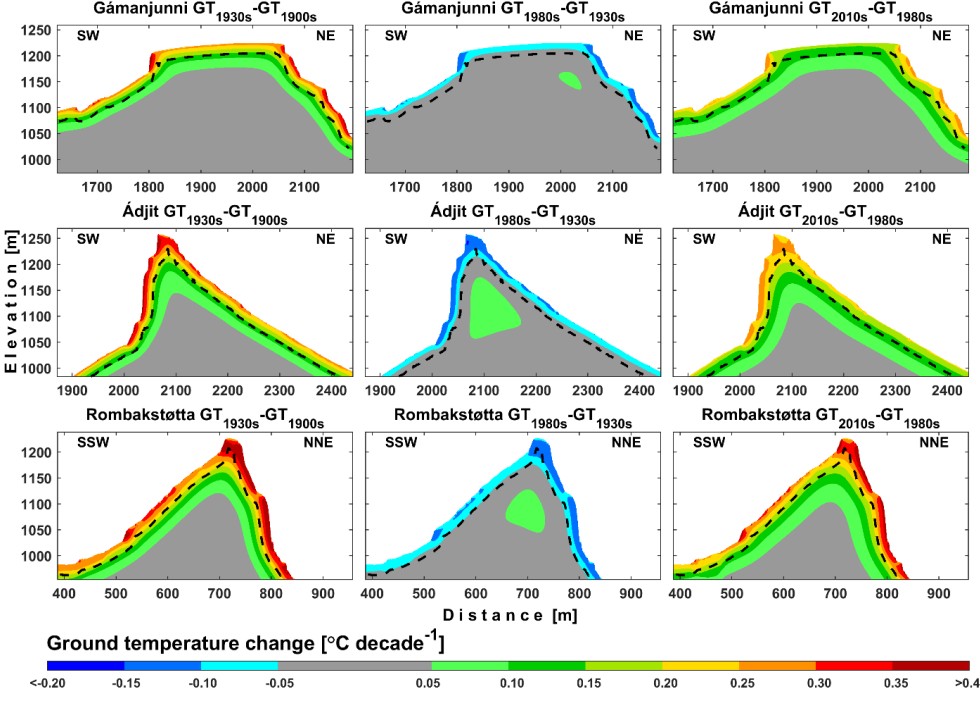


**Figure 9. Rate of change in 10 year mean GT for the profiles in the Northern Norway between the following decades: (1) the 1900s and the 1930s, (2) the 1930s and the 1980s, (3) the 1980s and the 2010s. 20 m depth is delineated by the black dashed lines.**





**Figure 10. GT in rock walls at 20 m depth simulated in the sensitivity scenarios for the 2010s. A cluster of values without any break elevation-wise usually represents a single rock wall.**





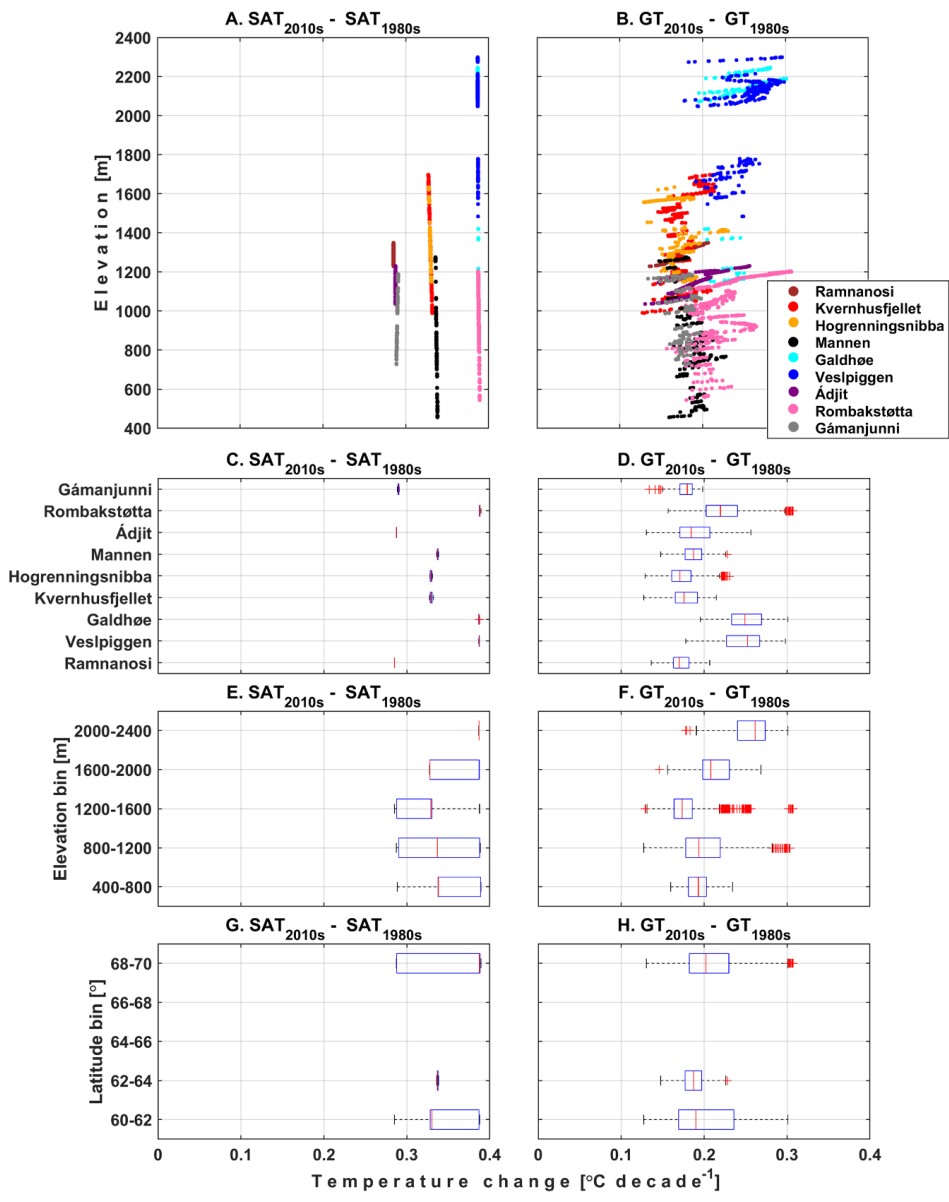

**Figure 11. Rates of SAT and GT change at approximately 20 m depth between the 1980s and the 2010s for all nodes below steep rock slopes (slope gradient > 60°). Lower subplots: Boxplots with SAT and GT rise between the 1980s and the 2010s for: (C-D) every profile, (E-F) 400 m elevation bins and (G-H) 2-degree latitude bins.**






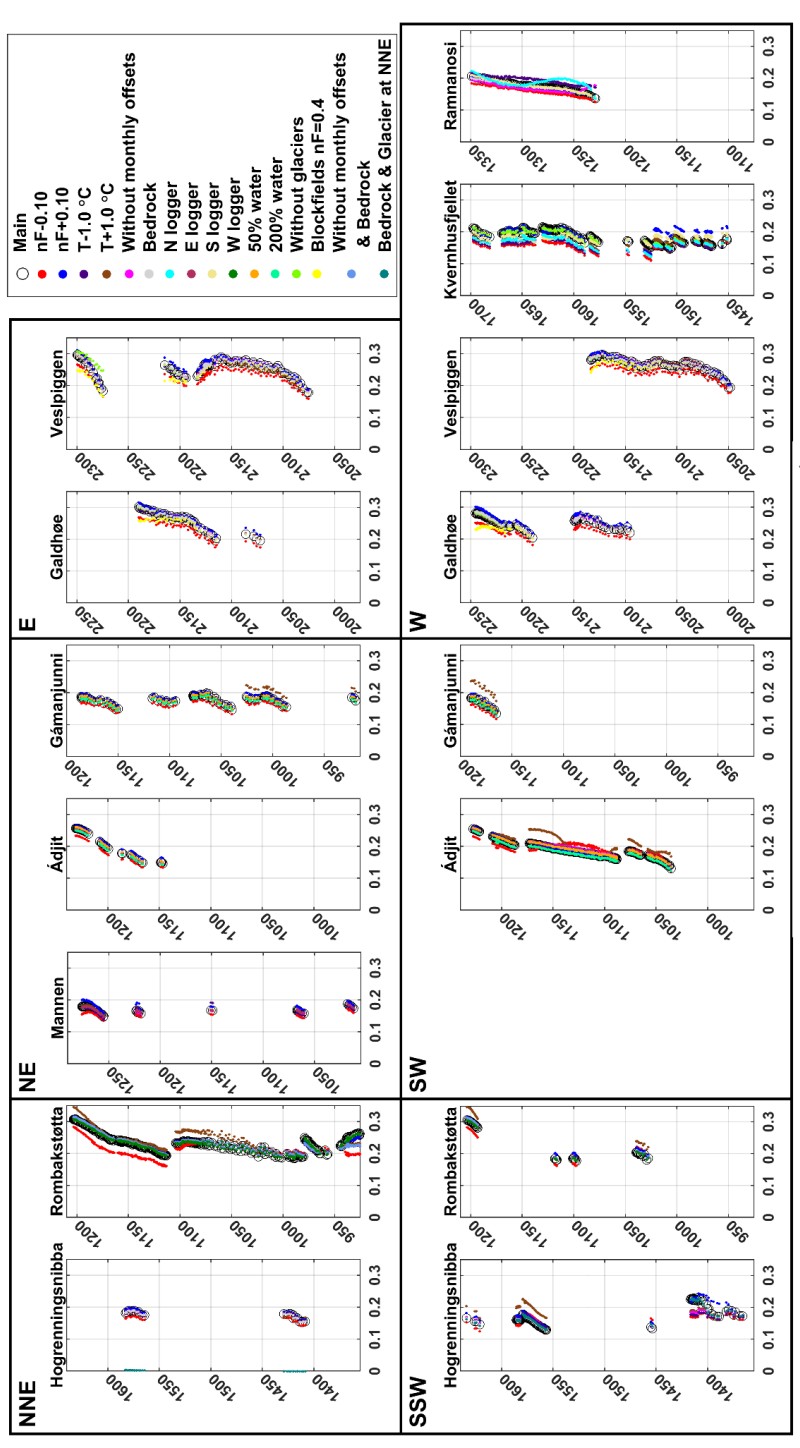

**Figure 12. Rates of GT change in rock walls at 20 m depth simulated in the sensitivity scenarios between the 1980s and the 2010s. A cluster of values without any break elevation-wise usually represents a single rock wall.**




**Tables**

**Table 1. Assumed depths of subsurface layers, along with volumetric fractions of the soil constituents for each layer: $\theta_w$ – volumetric water content; $\theta_m$ – volumetric mineral content; $\theta_o$ – volumetric content of organic matter; $\theta_a$ – volumetric air content; $z$ – depth. All sediment classes are underlain by bedrock with the**
**same ground composition as bedrock class ("a").**

| $z$ [m] | $\theta_w$ [-] | $\theta_m$ [-] | $\theta_o$ [-] | $\theta_a$ [-] |
|---|---|---|---|---|
| **"a": Bedrock (NGU code 130)** | | | | |
| >0.0 | 0.05 | 0.95 | 0.00 | 0.00 |
| **"b": Thin till (NGU code 12);** | | | | |
| **"c": Thin colluvium (NGU code 82)** | | | | |
| 0.0–1.0 | 0.30 | 0.60 | 0.00 | 0.10 |
| **"d": Medium thick till** | | | | |
| 0.0–1.0 | 0.30 | 0.60 | 0.00 | 0.10 |
| 1.0–2.0 | 0.40 | 0.60 | 0.00 | 0.00 |
| **"e": Thick till (NGU code 11);** | | | | |
| **"f": Thick colluvium (NGU code 81)** | | | | |
| 0.0–2.0 | 0.30 | 0.60 | 0.00 | 0.10 |
| 2.0–10.0 | 0.40 | 0.60 | 0.00 | 0.00 |
| **"g": Weathered material** | | | | |
| 0.0–2.0 | 0.10 | 0.60 | 0.00 | 0.30 |
| **"h": Thin organic cover over bedrock or shallow regolith (NGU code 100)** | | | | |
| 0.0–0.5 | 0.40 | 0.50 | 0.10 | 0.00 |
| **"i": Thin regolith (NGU code 72)** | | | | |
| 0.0–1.0 | 0.10 | 0.60 | 0.00 | 0.30 |
| 1.0–2.0 | 0.40 | 0.60 | 0.00 | 0.00 |

| $z$ [m] | $\theta_w$ [-] | $\theta_m$ [-] | $\theta_o$ [-] | $\theta_a$ [-] |
|---|---|---|---|---|
| **"j": Medium thick regolith** | | | | |
| 0.0–1.0 | 0.20 | 0.60 | 0.00 | 0.20 |
| 1.0–4.0 | 0.40 | 0.60 | 0.00 | 0.00 |
| **"k": Fluvial/Alluvial sediments (NGU code 50)** | | | | |
| 0.0–1.0 | 0.10 | 0.60 | 0.00 | 0.30 |
| 1.0–10.0 | 0.40 | 0.60 | 0.00 | 0.00 |
| **"l": Blockfields (NGU code 73)** | | | | |
| 0.0–2.0 | 0.10 | 0.60 | 0.00 | 0.30 |
| 2.0–5.0 | 0.40 | 0.60 | 0.00 | 0.00 |
| **"m": Rock glacier (NGU code 88)** | | | | |
| 0.0–2.0 | 0.05 | 0.60 | 0.00 | 0.35 |
| 2.0–5.0 | 0.10 | 0.60 | 0.00 | 0.30 |
| 5.0–35.0 | 0.40 | 0.60 | 0.00 | 0.00 |
| **"n": Scree** | | | | |
| 0.0–5.0 | 0.02 | 0.40 | 0.00 | 0.58 |
| 5.0–various depths | 0.60 | 0.40 | 0.00 | 0.00 |
| **"o": Fractured bedrock** | | | | |
| 0.0–10.0 | 0.05 | 0.80 | 0.00 | 0.15 |
| 10.0–various depths | 0.10 | 0.90 | 0.00 | 0.00 |
| **"p": Heavily fractured bedrock** | | | | |
| 0.0–10.0 | 0.05 | 0.75 | 0.00 | 0.20 |
| 10.0–various depths | 0.15 | 0.80 | 0.00 | 0.05 |
| **"q": Very thick colluvium** | | | | |
| 0.0–2.0 | 0.05 | 0.60 | 0.00 | 0.35 |
| 2.0–5.0 | 0.10 | 0.60 | 0.00 | 0.30 |
| 5.0–30.0 | 0.40 | 0.60 | 0.00 | 0.00 |





Table 2. SAT records used to construct forcing along profiles.

| Mountain, municipality | Meteorological station at the lower elevation along the profile (elevation; years with records) | Meteorological station on the mountain plateau (elevation; years with records) | Meteorological station(s) with the long-term temperature records (elevation; years with records) |
|---|---|---|---|
| **Western Norway** | | | |
| Mannen, Rauma | Marstein (67 m; 2010–present) | Mannen (1294 m; 2010–present) | Bergen-Lungegårdshospitalet (17 m; 1861–1895); Bergen-Pleiestiftelsen (22 m; 1895–1926) |
| Hogrenningsnibba, Stryn | seNorge (200 m; 1957–present) | seNorge (1600 m; 1957–present) | |
| Kvernhusfjellet, Stryn | | | |
| Ramnanosi, Aurland | seNorge (40 m; 1957–present) | Klevavatnet (960 m; 2014–present) | |
| **Jotunheimen** | | | |
| Veslpiggen, Lom | Juvvasshøe (1894 m; 1999–present) | seNorge (2230 m; 1957–present) | Dombås II (643 m; 1864–1972) |
| Galdhøe, Lom | | | |
| **Northern Norway** | | | |
| Gámanjunni 3, Kåfjord | seNorge (250 m; 1957–present) | Gámanjunni (1237 m; 2016–present) | Tromsø I (38 m; 1872–1926) |
| Ádjit, Storfjord | Skibotn II (20 m; 2004–present) | | |
| Rombakstøtta, Narvik | Straumsnes (200 m; 2011–present) | Narvik-Fagernesfjellet (1000 m; 2014–present) | |






**Table 3. Assumed nF-factors along the profiles, which depend on the slope gradient.**

| Slope gradient [°] / Sediment or vegetation class | nF-factor | | |
|---|---|---|---|
| | **Western Norway** | **Jotunheimen and Rombakstøtta** | **Gámanjunni and Ádjit** |
| <30 | 0.25 | 0.40 (based on data from Gisnås et al., 2014) | 0.50 (based on data from Eriksen, 2018b) |
| 30–40 | 0.50 | 0.55 | 0.60 |
| 40–50 | 0.70 | 0.70 | 0.75 |
| 50–60 | 0.90 | | |
| >60 | 1.00 | | |
| Blockfields (Jotunheimen) | | 0.70 (PACE, BH-1 and BH-2) | |
| Rock glacier (Ádjit) | | | 0.80 (based on data from Eriksen, 2018a) |
| Broad-leaved forest | 0.25 (Gisnås et al., 2017) | | |





**Table 4. Summary of the exposures for the rock walls along profiles. The direction measuring system with respect to the north azimuth is used. "Easternmost" - aspects between 0°–180°; "Westernmost" - aspects between 180°–360°.**

| Mountain, municipality | Main profile aspect of the westernmost rock wall [°] | Logger data for the first rock wall | Main profile aspect of the easternmost rock wall [°] | Logger data for the second rock wall |
|---|---|---|---|---|
| Mannen, Rauma | None | | 38 | Two runs: N (350°) as the main run and E (90°) |
| Hogrenningsnibba, Stryn | 200 | S (210°) | 20 | N (320°) |
| Kvernhusfjellet, Stryn | 272 | Three runs: W (270°) as the main run, N (320°) and S (210°) | None | |
| Ramnanosi, Aurland | 271 | Three runs: W (280°) as the main run, N (10°) and S (220°) | None | |
| Veslpiggen, Lom | 294 | W (297°) | 85 | Eh (89°) |
| Galdhøe, Lom | 270 | W (297°) | 68 | El (82°) |
| Gámanjunni 3, Kåfjord | 260 | Two runs: S (200°) as the main run and W (320°) | 80 | N (360°) |
| Ádjit, Storfjord | 228 | Sh (190°) | 48 | N (30°) |
| Rombakstøtta, Narvik | 202 | Two runs: E (100°) as the main run, because the west-facing logger is too cold, and W (270°) | 37 | N (25°) |



**Table 5. Sensitivity scenarios.**

| Scenario(s) | Modifications | Profiles |
|---|---|---|
| "nF-0.1"/ "nF+0.1" | We modify nF-factors by subtracting 0.1 or adding 0.1. | All |
| "T-1 °C"/ "T+1 °C" | We subtract or add 1 °C to the forcing data before applying nF-factors. | |
| "Without monthly offsets" | We ignore solar radiation and force the model directly with SAT; however, we still account for the nival offsets. | |
| "N/E/S/W logger" | We test thermal influence of SOs measured in the other rock wall aspects as listed in Table 4. | Mannen, Kvernhusfjellet, Ramnanosi, Gámanjunni and Rombakstøtta |
| "50 % water"/ "200 % water" | The water fraction is reduced by 50 %/increased by 200 % compared to the values in the main run and the remaining fraction is added to/subtracted from the mineral fraction. | Gámanjunni and Ádjit |
| "Bedrock" | We assume that the entire subsurface is composed of the bedrock. | Ramnanosi, Hogrenningsnibba, Veslpiggen, Galdhøe and Rombakstøtta |
| "Without glaciers" | We remove glaciers and perennial snow along profiles. | Galdhøe, Veslpiggen and Kvernhusfjellet |
| "Blockfields nF=0.4" | We change nF-factor for blockfields to 0.4. | Galdhøe and Veslpiggen |
| "Snow patch" | At Hogrenningsnibba snow persisted until late summer in some years, hence we add a snow patch on the top of the mountain and partly along the northern-facing slope. | Hogrenningsnibba |
| "Bedrock & Glacier at NNE" | We test what happens if Hogrenningsnibba has no sediments and add a glacier at the NNE-facing slope. | |
| "Without monthly offsets & Bedrock" | We remove monthly surface offsets and assume that the subsurface consists only of bedrock. | Rombakstøtta |




**Videos**

In the current version of the manuscript, videos are available through the University of Oslo's OneDrive account:
https://uio-my.sharepoint.com/:f:/g/personal/justync_uio_no/EjO_zEqsoixAju0-
h1198IgBbru2nFgngZuyDb0tl9KeMQ?e=dzmVrA . Note that the file is view-only. The videos can be viewed
directly in any web browser, except for Internet Explorer 11, or downloaded (file size is 124 MB).