# Peer review of "Post Little Ice Age rock wall permafrost evolution in Norway"

_The Cryosphere, 2022_

## Author Comment (AC1)

**tc-2022-4**

**Post Little Ice Age rock wall permafrost evolution in Norway**

Justyna Czekirda, Bernd Etzelmüller, Sebastian Westermann, Ketil Isaksen, Florence Magnin

**General response to referees**

We would like to thank the two referees for their reviews. We modified the manuscript according to the comments made by the referees. We shortened Section 2, rewrote Section 4 and Section 5.2. We also changed some figures and tables, and reduced the number of figures and tables. Furthermore, we added a new Section 5.4 about the geomorphological implications and added a few supplementary figures and tables. We also asked our British colleague to proofread the manuscript.

**Response to individual referees**

| Anonymous Referee #1 | page 2  |
|----------------------|---------|
| Anonymous Referee #2 | page 11 |

**Referee comments are in bold**

Author replies are in normal text (*Italic is for the manuscript text*)

**Anonymous Referee #1**

**General Comments**

Czekirda et al. present an interesting analysis to describe the evolution of permafrost in rock walls over the last 100+ years. The analysis is based on 2-D thermal modelling and measurements from rock wall temperature loggers along 9 profiles in Norway. The authors consider the various factors influencing the permafrost distribution and its evolution in the simulations and also assess their relative importance. The results of the simulations indicate an increase in ground temperatures since the 1980s and that the rate of change increases with elevation within a single rockwall section. Overall the analysis, interpretation of results and conclusion appear to be sound. The paper is for the most part, well written but some editorial revisions are required. The paper would be of interest to the general permafrost research community and especially those interested in mountain permafrost distribution and the stability of rockwalls. The paper is therefore worthy of publication following minor revision. I have a number of comments for the authors' consideration.

We want to thank Anonymous Referee #1 for the constructive review, together with many detailed suggestions how to correct our grammatical errors and improve the language. We also extended the model validation and discussion of our results in view of the borehole data and geophysical surveys as suggested by Anonymous Referee #1.

It is good that field measurements such as the rock wall temperature measurements have been utilized in model calibration. However, there is not much comparison of the simulated results with observations. There is some comparison of the data (near-surface temperatures) derived from the rock wall loggers with the simulated temperatures but as the authors point out, these data were used for the calibration of the forcing input and only qualitative comparisons are done. There is not much comparison to observations of deeper temperatures, although a comparison to one borehole is mentioned. There appears from the information in section 2 and the references cited there are a number of boreholes with temperature cables (e.g. Christiansen et al. 2010) in the study area and also geophysical surveys have been done (e.g. Etzelmuller et al. 2021). It is not clear whether any of the boreholes or geophysical surveys are located on or close to the study transects. It would be good if the paper could include a comparison of these observational data with the simulated results. This could help strengthen discussion with respect to the relative importance of forcing factors (and limitations of the model) and other modes of heat transfer such as convection or advection along the various profiles.

We thank Anonymous Referee #1 for the valuable comments about model validation. We added model validation based on the borehole data and geophysical surveys for Galdhøe, Mannen and Gámanjunni.

A. Borehole data: We can compare the profile at Galdhøe to the borehole data, because other boreholes in Norway are too far from our profiles. We added Supplementary Figure S24 and Supplementary Table S5, where we compare simulated ground temperatures to borehole data (e.g. Christiansen et al. 2010) in the gentle terrain in the Jotunheimen Mountain Range. The borehole locations are added to the overview map in Figure 1. We added "*Supplementary Figure S24*. *Comparison of annual modelled*

ground temperature at depths below 5 m along Galdhøe profile to measured ground temperatures by 100 m deep PACE borehole in the Jotunheimen Mountains. RMSE-root mean square error, MAE-mean absolute error, ME-mean error. Refer to Figure 1 in the main manuscript for the borehole location.", which shows R2=0.75, RMSE=0.08, MAE=0.06, ME=0.03. Other boreholes in the Jotunheimen Mountain Range are shallower, therefore we only compare modelled and simulated ground temperature at 10 m depth for all boreholes over the 2010s ("Table S5. Comparison of simulated ground temperatures at the Galdhøe profile and measured ground temperatures in the Jotunheimen Mountains over the 2010s. Refer to Figure 1 in the main manuscript for the borehole locations"). The latter table includes three snow scenarios ("Main", "nF+0.1", "nF-0.1") and the absolute difference between modelled and measured ground temperature is maximum 0.52 °C. The compared period length and depths are justified by the study focus on the inter-decadal variations in the ground temperature. We referred to the figure and table in the main manuscript by adding it to the following sentence:

"Results from thermal simulations, both the modelled GTs and deeper warming rates, are in good agreement with the available borehole data in the Jotunheimen Mountains (Supplementary Figure S24 and Supplementary Table S5), although there are variations in snow conditions between the boreholes, hence we compared the measurements to various snow sensitivity simulations."

B. Geophysical investigations: We extended a comparison of our results to the geophysical surveys as suggested by Anonymous Referee #1. Geophysical investigations can be used to evaluate the main patterns of the modelled subsurface thermal fields for Mannen, Gámanjunni and at the lower permafrost limit along the Galdhøe profile. Etzelmüller et al. (2022) compared geophysical surveys, using the 2D and 3D electrical resistivity tomography (ERT) and/or refraction seismic tomography, to the modelled subsurface thermal fields for Mannen and Gámanjunni. We referred to their paper in our manuscript. We are of course aware that the conductive thermal field is perturbed by the non-conductive heat transfer mechanisms in larger fractures. This has been discussed in more detail in Etzelmüller et al 2022 for the Gámanjunni and Mannen. The thermal modelling shows the overall development of the thermal conditions over more than a century, while the ERT and seismic analysis give a detailed local snapshot of resistivity, which only partly can be transferred into a thermal signal. However, at Gámanjunni the overall pattern can be reproduced, both in relation to temperature differences due to topographic aspect and possible lower limits of permafrost in the slope.

We add the following text about Gámanjunni to the manuscript (Subsection 5.2.3):

"The results shown in Etzelmüller et al. (2022) for Gámanjunni show a somewhat different subsurface GT field, due to different model forcing. However, geophysical surveys reproduce the main patterns of the modelled subsurface thermal field at Gámanjunni presented in our study, and in Etzelmüller et al. (2022). The geophysical surveys at Gámanjunni indicate: (1) the thermal influence of the NW and SW facing rock walls, (2) higher resistivity (i.e. cooler conditions) in the scree below the SW-facing rock wall, (3) a warmer subsurface below the snow-covered plateau. In comparison with Etzelmüller et al. (2022), our thermal fields show (1) and (2) agree even better with the geophysical surveys, because we accounted for the additional surface offsets in the SW-facing rock wall. The conductive thermal field is slightly perturbed by the non-conductive heat transfer mechanisms in larger fractures. Etzelmüller et al. (2022) argued that comparison of the modelled ground temperature

and geophysical surveys is useless at smaller scales, due to high resistivity variations in rough terrain, influenced by crack and fractures, strong topographic variations and local water infiltration."

**We add the following text about Mannen to the manuscript (Subsection 5.2.1):**

"At Mannen, both the geophysical surveys presented in Etzelmüller et al. (2022) and our thermal modelling suggest that the existence of discontinuous permafrost is unlikely. The geophysical data indicate that sporadic permafrost can occur in the Mannen area; however, high resistivity values (> 20 k $\Omega$ m) measured in this area could also reflect very good water drainage conditions, due to highly fractured bedrock or even ion-poor pore water (Dalsegg and Rønning, 2012)."

We move this sentence about geophysical surveys in Jotunheimen to Subsection 5.2.2:

"For the BH5 borehole in Jotunheimen (Figure 1D) and nearby gentle slopes, geophysical surveys performed in 1999 and 2010, together with numerical modelling, indicated the degradation of permafrost over the intervening decade (Isaksen et al. 2011)."

We proceed with the following text about geophysical surveys close to the Galdhøe profile:

"We compared the modelled subsurface thermal fields for Galdhøe to the geophysical surveys from 1999 and 2010, and our results show a similar pattern of possible permafrost degradation in this marginal permafrost area (Supplementary Figure S25). The results are especially similar for the sensitivity simulation with less snow ("nF+0.1")."

**Specific Comments**

Language revision and other minor suggestions on the following lines: L17-18, L22, L38, L85-86, L96, L98, L102, L127, L132-133, L135, L155, L156, L195, L197, L216, L239, L268, L270, L288-289, L301, L387-388, L464, L481, L484, L793, L800, L803, L805, L828, L840, L855, L1203-1208.

Done as suggested. See the revised manuscript.

Comments on the following lines: L145, L334, L339, L357, L381, L382, L383, L408, L441, L459, L465, L469, L709, L737.

We removed these parts due to the comments from Anonymous Referee #2.

**L33-35 – You could be clearer here that this was due to an extreme event rather than long-term change.**

We changed the sentence to:

"An example of a fast response was exceptional rockfall activity reported during the extremely hot summers of 2003 and 2015 in the European Alps, likely due to permafrost degradation (Gruber et al., 2004; Ravanel et al., 2017)."

**L40-45 – Could you refer to the 3D (or multi-dimensional?) nature of heat flow and the higher thermal conductivity of the rock (compared to soil, unconsolidated sediments) here.**

We rewrote the sentence to:

"Rock wall permafrost is highly susceptible to climate deterioration because: (1) small latent heat effects and high thermal conductivity cause more rapid ground temperature (GT) increase (Gruber and Haeberli, 2007), (2) the three-dimensional nature of heat flow leads to faster degradation of deeper permafrost in some locations than would be the case in flatter terrain (Noetzli et al., 2007), and (3) thermal conditions in steep bedrock and the atmosphere are strongly coupled since steep slopes are typically have shallow snow or surface material, if any (e.g. Boeckli et al., 2012; Myhra et al., 2017)."

High thermal conductivity of rock is a result of low ice content, so we modified (1) and added thermal conductivity.

**L53-56 – Aren't freeze-thaw cycles important to this process (thermal contraction/expansion)? Would the frequency of these cycles change with climate warming?**

Ice segregation is probably the most likely mechanism of frost weathering (Walder and Hallet, 1986), and the importance of freeze-thaw cycles was discussed in e.g. Hallet et al. (1991). Freeze-thaw cycles were merely important since they bring rock to the required temperature range for frost weathering (e.g. -6 to -3 °C) and more water is then available. We included also "*and/or volumetric expansion*" as possible process of frost weathering. Volumetric expansion in situ when water freezes to ice depends on freeze-thaw cycles. Therefore, freeze-thaw cycles are already included.

**References**

Hallet, B., Walder, J. S., and Stubbs, C. W.: Weathering by segregation ice growth in microcracks at sustained subzero temperatures: Verification from an experimental study using acoustic emissions, Permafrost Periglac, 2, 283-300, https://doi.org/10.1002/ppp.3430020404, 1991.

Walder, J. S. and Hallet, B.: The Physical Basis of Frost Weathering: Toward a More Fundamental and Unified Perspective, Arctic and Alpine Research, 18, 27-32, https://doi.org/10.1080/00040851.1986.12004060, 1986.

L76-78 – Are you referring to the BTS approach here (e.g. Hoelzle 1992 https://doi.org/10.1002/ppp.3430030212; Gruber and Hoelzle 2001 https://doi.org/10.1002/ppp.374; Bonnaventure and Lewkowicz 2008 https://doi.org/10.1139/E08-013)

No. In that paragraph we only address studies in rock walls.

**L89 – Why not just refer to limits of permafrost occurrence rather than near-surface permafrost.**

We referred to it as "*near-surface permafrost*" due to the limitations mentioned at the beginning of Subsection 5.2. The study of Magnin et al. (2019) is based on a statistical model and 1981-2010 surface temperature, thus the model does not account for e.g. lateral heat fluxes or deeper permafrost preserved from colder periods.

**L91-93 – How were the transects chosen – are they representative of the geological and climate conditions in the region?**

The profiles were chosen because we run thermal monitoring of rock walls at the sites, and the modelling profiles cross these rock walls. The selection of the monitoring sites was originally done due to monitoring work of unstable rock slopes by governmental authorities (Gámanjunni, Mannen, Ramnanosi) or the Geological Survey of Norway (NGU) measure displacement rates at these sites (Ádjit, Rombakstøtta). The profile over Galdhøe was chosen due to its vicinity to Galdhøpiggen and the variation between steep slopes, flat block fields and glacier coverage. Climate variation is ensured due to the spatial dispersion of the sites.

**L99 – revise to "mean air temperature" – is this "mean daily air temperature"? (note: important to be clear that this is air temperature since surface and ground temperatures are also mentioned in text). It would be useful to give the normal mean annual air temperature as well as the range, here and in description for the other study areas.**

We added "*mean monthly air temperature*". The value was computed based on the mean monthly air temperature.

We added to 2.1 Western Norway:

"Normal mean annual temperature (the normal period 1971-2000) varies between -5 - -4 °C at the highest mountain peaks to 6 - 8 °C in the coastal areas (Lussana, 2020) and the annual range of mean monthly air temperature is less than 18 °C (Tveito et al., 2000)."

We added to 2.2 Jotunheimen:

"Normal mean annual air temperature (1971-2000) is under -6 °C at the highest mountain peaks to 0-2 °C in the valleys (Lussana, 2020). The area has an annual range of mean monthly air temperature normally greater than 18 °C (Tveito et al., 2000)."

We added to 2.3 Northern Norway:

"Normal mean annual air temperature (1971-2000) is between -6 - -5 °C at the highest mountains to 2-6 °C in the coastal areas (Lussana, 2020)."

L103 – Revision suggested: "...2015-2017 nine loggers have been installed at selected rock walls to measure surface temperature in western Norway". Since the data from these loggers appears to be used in your study, you should probably mention the type of logger and its accuracy and precision.

We add the accuracy already in the introduction, since the accuracy applies to all loggers:

"Systematic field observations using Geoprecision, M-Log 5W Rock rock wall loggers (at least 0.1 °C at 0 °C accuracy) were taken at selected sites in the Jotunheimen Mountains (Hipp et al., 2014)."

**L136–144 – Are these boreholes and geophysical surveys on or near the study profile?**

Yes, the boreholes are quite close to the Galdhøe profile. We added borehole locations to Figure 1D and added the following text to the figure caption:

"Black circles show borehole locations."

Furthermore, we refer to the figure in the text about boreholes:

"In 1982, the first 10 m deep borehole at 1851 m elevation was drilled in Jotunheimen (Ødegård et al., 1992) and then in August 1999, the deepest permafrost borehole (129 m) in Norway was drilled in the continuous permafrost zone at Juvvasshøe (1894 m) as part of the PACE project (Figure 1D; Sollid et al., 2000; Harris et al., 2001). Additional boreholes have been drilled at various elevations in the Juvvasshøe area on its north-eastern slope in August 2008 (Figure 1D; Farbrot et al., 2011)."

The geophysical surveys were conducted next to the borehole BH5, which is not so far from the Galdhøe profile. We already mentioned the comparison under the general comments above.

L140-143 – Do you mean that permafrost occurs at least at elevation as low as 1559 m and that frozen conditions exist at all the boreholes down to this elevation? OR is it present at some boreholes but not others? Some clarification and revision of text required.

We rewrote "The measured GTs show that discontinuous permafrost occurs down to at least the borehole drilled in the bedrock at 1559 m elevation" to "The measured GTs show that permafrost occurs in all boreholes at and above 1559 m elevation."

**L159-160 – Were these boreholes on or near the study profiles?**

No, they are not so near the study profiles. We decided to mention the closest ground temperature measurements to Gámanjunni presented in the study of Blikra and Christiansen (2014) in Subsection 5.2.3. That study derived the local discontinuous permafrost limit at an elevation around 700 m for the Jettan rockslide area, around 12 km north-west from Gámanjunni. We add the following sentence to Subsection 5.2.3.:

"Furthermore, the local permafrost limit at an elevation of around 700 m, derived from various temperature measurements at the Jettan rockslide (Blikra and Christensen, 2014), 12 km NW of Gámanjunni, is in accordance with our modelled permafrost limit for less sun-exposed slopes."

**References**

Blikra, L. H. and Christiansen, H. H.: A field-based model of permafrost-controlled rockslide deformation in northern Norway, Geomorphology, 208, 34-49, https://doi.org/10.1016/j.geomorph.2013.11.014, 2014.

**L216 – Do you mean "surficial deposits"? also "all of" might be better than "the entire"**

We changed it to "surface material". We changed "the entire" to "all of".

**L249-251 – How far back before 1900 do you reconstruct SAT?**

We reconstruct SAT as far back as we can, i.e. for measurement periods mentioned in Table 2 (1861 – Western Norway, 1864 – Jotunheimen, 1872 – Northern Norway).

We added a sentence to clarify it:

"The latter data allows for SAT reconstruction back to 1861 for Western Norway, 1864 for Jotunheimen and 1872 for Northern Norway."

**L259 – revise to "GST determined from rock wall loggers". It isn't clear what you mean by giving more reliability as SAT and GST are not the same thing and there are offsets which you mention later in the paper.**

Rock wall loggers are installed in locations, which should be snow free. Therefore, we expect little differences between SAT and GST in e.g. December, when it is almost dark or polar night at our study sites and the influence of shortwave solar radiation is minimal. If offsets between SAT and GST in December are considerable, it is likely that there are some errors in our procedures to reproduce SAT, unless we see some influence of snow. We also compared the loggers at individual sites, since they should have similar temperatures during e.g. December. Therefore, we think that comparison of SAT and GST evolution in rock wall loggers for each site gives more reliability. We rewrote the sentence, so it is clearer what we meant:

**"This allows for a comparison of SAT with GST determined from rock wall loggers in months with minimal shortwave radiation, e.g. December, and gives more reliability."**

**L455-458 – I may have missed this, but do you mention why you chose these time periods? I assume it is because they coincide with periods of warming and cooling in the SAT record. Also, be clear that you are referring to simulated GT.**

We did not mention why we chose these time periods. We chose these periods as the most crucial for GT evolution after the Little Ice Age. It is well known from other studies that the 1930s was a warm decade and the 1970s/1980s was a cold period around Scandinavia. SAT variations are shown in "Appendix A. Surface air temperature trends".

We added "*simulated/modelled*" in front of most "GT(s)" in Subsection 4.3.

**L545 – Shouldn't you mention that the ice content is important?**

That is a good point. We added the following sentence:

"However, warming retardation due to the latent heat effects depends on the ice content, and results from the assumed 5 % vol. ice content for fully frozen ground, thus for lower ice contents, latent heat effects are smaller."

L641 – Something else to consider in this section. Surface temperature in winter and therefore nF will not just be a function of the snow depth but will also depend on active layer thickness and substrate conditions (especially moisture content) as these will influence the latent heat effect (see for example Riseborough and Smith 1998, 7th Int. Permafrost Conf. Proc.; Throop et al. 2012 doi:10.1139/E11-075).

We added the factors mentioned in Throop et al. (2012) to this sentence:

"The overall effect of snow cover on GT is complex because it depends on snow thickness, depth, duration, timing, melting processes within a snowpack, snow structure (Zhang, 2005), sun exposure (Magnin et al., 2017b), MAAT, substrate, the thickness of the active layer and ground moisture (Throop et al., 2012)."

L663-665 – Late-lying snow cover would delay or reduce the spring warming of the ground. Also, latent heat required to melt snow reduces amount of heat available to heat the ground.

We rewrote the sentence to:

"Snow cover increases albedo of the surface and thus reduces absorbed short-wave radiation, meaning late-lying snow would delay or reduce the spring warming of the ground (e.g. Hasler et al., 2011a; Magnin et al., 2017b)."

Those studies do not mention much about latent heat effects, which we already covered here based on another study:

"(4) Snow requires large energy inputs to melt, hence GT will be lower than SAT during snowmelt; however, this usually lasts for a short time and may be unimportant on annual time scales (Zhang, 2005)."

L675-672 – Effect of thin or late onset of snow accumulation is also shown by Palmer et al. (2012 doi:10.1139/E2012-002). The temporary ground cooling related to low snow cover is also reported for the European Alps by PERMOS (2019) and Noetzli et al. (2020 https://doi.org/10.1175/BAMS-D-20-0104.1)

We extended point (5) as suggested:

"(5) During autumn, thin snow cover could lead to an enhanced conductive heat flux from the ground due to large thermal gradients between the cooled snow surface and warmer upper ground layers (Keller and Gubler, 1993; Luetschg et al., 2008). Furthermore, in the low-snow years, GT at the top of permafrost is relatively constant during freezeback and may be higher than GST that is coupled to SAT, leading to positive thermal offsets (Palmer et al., 2012). In addition, temporary ground cooling was observed at several sites across Switzerland during one or two winters in 2015-2017, when snow cover arrived very late and was thinner than usual (PERMOS, 2019; Noetzli et al., 2020). The latter cooling effect was not recorded at steep bedrock sites, where GT is usually insensitive or less sensitive to snow cover changes (PERMOS, 2019; Noetzli et al., 2020)."

This also emphasises the strong coupling between SAT and GST in rock walls, thus little sensitivity to snow trends in rock walls.

L705-707 – Are you referring to the thermal offset here which is related to the difference between frozen and unfrozen thermal conductivity (as well as the lag effects that mean that permafrost can still be present at depth when surface temperatures are above  $0^{\circ}$ C).

No, we do not mention thermal offsets related to the difference between frozen and unfrozen thermal conductivity at all. Thermal offsets due to seasonal variations in thermal conductivity are quite small in bedrock due to low porosity.

We refer to temperature offset in the previous sentence:

"(1) Neither the rock wall temperatures nor the statistical modelling account for the temperature offsets deeper in the ground, e.g. measurements conducted by Hasler et al. (2011a) in the European Alps were even 3 °C lower at depth than mean annual rock surface temperatures, hence the existing surface information might be insufficient". This offset is according to the authors likely due to late-lying snow cover and ventilation effects.

**L733-735 – It would be useful to show this comparison, maybe in an appendix or supplementary information.**

Done. See the general comments above.

**L788-789 – Are these rates determined for the same time period?**

No, the periods were not the same. Rock wall rate was average over the 1980s-2010s (results shown in the article) and Juvvasshøe borehole data was since 1999. We only wanted to compare roughly our warming rates to borehole data. We added the periods to emphasise the difference:

"The modelled warming rate of on average 0.25 °C decade-1 in rock walls in Jotunheimen over the 1980s-2010s is slightly higher than the warming rate of 0.2 °C decade-1 measured at 20 m depth in the deep borehole at Juvvasshøe since 1999 (Smith et al., 2021)."

We do not have borehole data before 1999 and we want to keep warming rates in rock walls between the 1980s and the 2010s as one of the foci in the paper, since changes occurred already since the 1980s.

**Anonymous Referee #2**

We thank Anonymous Referee #2 for his/her comments, and we try to address all comments in the following.

The work presents results from 2D numerical modelling of heat conduction from several transects crossing steep rock walls in Norway. The aim of the study is not explicitly stated (it needs to be), but 1.92-93 and the choice of sites suggest the aim is to investigate bedrock temperatures behind steep rockwalls and the adjacent landscape, including at or near the sites of known slope instabilities. At least one transect crosses an active instability (Gámanjunni).

In the revised version, we have clarified the goals. The section with the goals reads now:

"The aim of this study is to improve knowledge about the spatio-temporal variations in ground temperature in steep rock walls in Norway on the inter-decadal scale. We employ the 2D slope-scale transient heat flow model CryoGrid 2D (Myhra et al., 2017) to simulate the thermal evolution of mountain permafrost since 1900 along nine transects crossing the instrumented rock walls in mainland Norway. We advance the methods presented in the study by Myhra et al. (2017), by an observation-constrained model for ground surface temperature (GST), i.e. including the field observations from rock walls in various expositions. All sites presented in this study are monitored by at least one rock wall logger in a vertical rock face, and three of the unstable sites are constantly monitored by the Norwegian Water and Energy Directorate (NVE). Thus, this study aims to be an important baseline for the development of the ground thermal regime in steep mountain terrain, which is possibly unstable."

Gámanjunni is only one of several instabilities. Gámanjunni, Mannen and Ramnanosi are under constant surveillance by NVE (The Norwegian Water and Energy Directorate). Others are under observation by the Geological Survey of Norway (NGU)/ the University of Oslo (UiO) in terms of GPS measurement of movement (the Loen sites, Rombakstøtta), while only the Jotunheimen sites do not have documented instabilities. In contrast, there we have one of our permafrost observatories, which gives us a unique possibility to compare rock slope and gentle permafrost areas.

Reliably modelling bedrock temperatures down to depths of relevance for the evolution of slope instabilities where permafrost may exist is very important and within the scope of TC. However, it seems that the model used has never been validated against borehole measurements even in simpler topographic settings and thermal systems. No borehole data for validation is presented here either.

We validated the model against borehole data. See our replies to General Comments from Anonymous Referee #1.

By its formulation, the model cannot account for effects that are especially important in landscapes susceptible to slope instability, where more pervasive and widening fractures are to be expected.

We believe this could be a misunderstanding. The purpose of this study is not to present a detailed numerical modelling taking into account all processes in a steep slope with all cracks

etc., which is nearly impossible given the long temporal scale we chose. Neither we want to explicitly model slope instabilities. The purposes are to study the long-term evolution of ground thermal regime, and relate this to different regions, elevations and climate conditions in Norway, including permafrost. Such long-term modelling is not capable to include e.g. advective heat flow in cracks or large process space, this is the challenge for process models evaluating energy balance regimes over much smaller spatial scale or shorter time periods. So, no, this manuscript never tried to account for "landscapes susceptible to slope instability where more pervasive and widening fractures are to be expected".

**Snow is also treated in a very simple way, and several parts of the transects are not oriented favourably to the topography for a 2D model.**

Yes, snow is treated simple due to the same reasons discussed above. There are several studies clearly demonstrating that simple treatment of snow produces reliable conclusion when operating over large spatial or temporal scales. However, we agree that on a point or site scale, more sophisticated snow treatment should be used, as e.g. demonstrated in other studies from Norway and Svalbard (e.g. Schmidt et al., 2021; Zweigel et al., 2021).

**References**

Schmidt, J. U., Etzelmüller, B., Schuler, T. V., Magnin, F., Boike, J., Langer, M., and Westermann, S.: Surface temperatures and their influence on the permafrost thermal regime in high-Arctic rock walls on Svalbard, The Cryosphere, 15, 2491–2509, https://doi.org/10.5194/tc-15-2491-2021, 2021.

Zweigel, R. B., Westermann, S., Nitzbon, J., Langer, M., Boike, J., Etzelmüller, B., & Vikhamar Schuler, T.: Simulating snow redistribution and its effect on ground surface temperature at a high-Arctic site on Svalbard. Journal of Geophysical Research: Earth Surface, 126, https://doi.org/10.1029/2020JF005673, 2021.

**The resulting uncertainties are impossible to quantify through sensitivity experiments without real borehole data.**

We disagree here. We will never have boreholes in steep rock walls. Such boreholes exist in very, very rare cases. This means our sensitivity analyses indicate a range where the truth may be situated. We included model validation for the gentle terrain, which is in good agreement with our modelling.

Boreholes data suitable for comparison do exist, and are claimed to agree 'quite well' with the model. But they are not shown. If the current level of detail in the presentation and discussion of the model results is to be preserved, the validation against at least this set of boreholes needs to be included. The good agreement would build some confidence in the model performance.

This is a very valid comment, and we have addressed this issue. Anonymous Referee #1 had similar comments, so they are described in detail above.

Otherwise I suggest to significantly shorten the ms. and only focus on discussing those feature of the modelled bedrock temperatures that are broadly consistent across different sites or parts of different transects featuring similar conditions.

We agree, especially the results part was lengthy, and we have rewritten this part. We have especially avoided redundancies, and simplified and reduced the number of figures, and moved some figures and tables to Supplementary Figures and Tables.

I am particularly concerned because one of the cited papers (Kristensen et al. 2021) already applied the same model, again without borehole measurements, to an unstable slope that it described as highly fractured and with seasonally varying permeability. As such, a site completely unsuitable for a model only accounting for heat conduction. We still know very little about the links between temperatures and slope stability, and false leads from unproven models can cause unnecessary confusion. More details on the limitations of the model and why some of the broadly accepted assumptions would not apply here are below in my comments to the text.

The paper of Kristiansen et al. (2021) has undergone a rigorous review process and revision, is accepted and published in the well-accepted journal "Landslides". The paper does not deal with the Mannen instability at all, just a little part known as "Vesle-Mannen". Our model was used to give an impression about possible thermal regime development in this small slab, which failed eventually. Thus, the comment is irrelevant to our study.

We do not discuss any mechanical aspects at the study sites specifically, and we do not focus on rockslides in this study. To be clearer, we made some changes to the manuscript:

- We removed this sentence: "These results are a prerequisite for stability assessment in the Norwegian rock walls subjected to permafrost conditions.", which may be confusing.
- 2) We moved one paragraph about mechanical aspects from Introduction to a new Subsection "5.4 *Geomorphological implications*":

"Our study focuses on rock wall permafrost evolution in Norway since the end of the Little Ice Age. The results indicate a substantial increase of GT at 20 m depth since the 1980s at all sites in Norway. Although mechanical aspects of GT increase are not considered in our modelling, the ground thermal regime itself has an important influence on geomorphological processes in periglacial regions (e.g. Berthling and Etzelmüller, 2011) and ultimately landscape development (e.g. Egholm et al., 2015). The ground thermal regime and its temporal development in steep slopes certainly influence the weakening of rock bonds, widening of cracks and the potential for frost weathering processes. Several authors have linked permafrost degradation and destabilisation of slopes (e.g. Davies et al., 2000; Davies et al., 2001; Gruber and Haeberli, 2007; Krautblatter et al., 2013). Conductive warming of ice-filled fractures, which are believed to stabilise permafrost-underlain mountains (e.g. Dramis et al., 1995), may result in: (1) loss of joint bonding and reduction of shear strength of the joint due to water release through ice melting, (2) shear strength changes due to mechanical ice properties that are a function of the normal stress and temperature (Davies et al., 2001). *Furthermore, advective heat transport by percolating meltwater may result in rapid,* local degradation of rock wall permafrost, which can trigger rockfalls even in cold permafrost areas (Hasler et al., 2011b). Krautblatter et al. (2013) noticed, in addition, that rock-mechanical properties themselves depend on rock temperature; hence, thawing can lead to a significant drop in rock strength. Frost weathering processes caused by ice segregation and/or volumetric expansion of in situ water are believed to contribute to the generation of weakness planes or widening fractures in frost-affected rocks (Gruber and Haeberli, 2007; Krautblatter et al., 2013). It is uncertain how the modelled GT distribution and GT increase may affect slope stability. Our results suggest

that GT increase increases with elevation within a single rock wall section, hence this may indicate that instability risk is larger with elevation for a single rock wall section; however, GT may be highest in the middle of the rock wall, hence this part may be more susceptible to permafrost degradation in the sun-exposed rock walls. Furthermore, shaded rock walls may act as "refrigerators" in the landscape due to low snow cover within the rock walls and small amounts of solar radiation (e.g. Myhra et al., 2017). Thus, these landscape areas are locations for steep thermal gradients on the transition of snow-free steep rock walls and snow-covered more gentle terrain or glaciers/snowfield covered areas. This is exemplified in other studies and formerly addressed by *Myhra et al.* (2019) for the upper parts of talus slopes or rock glaciers below shaded rock walls, for cirques (Sanders et al., 2012) and below coastal cliffs in Arctic settings (Ødegård and Sollid, 1993; Wangensteen et al., 2007; Schmidt et al., 2021). All these settings influence frost weathering, as these strong thermal gradients favour frost segregation and frost cracking (Hales and Roering, 2007). Similar processes are also discussed for snow patches in relation to nivation processes (Berrisford, 1991). Thus, especially the constant change of ground thermal regime associated with rock walls and their vicinity are hot spots in material production and further geomorphological transport processes."

We mention here explicitly that: "*It is uncertain how the modelled GT distribution and GT increase may affect slope stability.*"

The ms. is not suitable for publication in its present form but see below for comments and suggestions that may make it acceptable after major revisions.

Throughout the text: improve the use of articles, e.g. l. 58 'the Northern Norway' -> 'Northern Norway', l. 100 'Permafrost limit' -> 'The permafrost limit', and in general improve the use of the English language and remove repetitions.

We improved the language.

**Specific Comments**

Language revision and other suggestions on the following lines: l. 67, l. 243, l. 247, l. 803, fig.1, fig.3, fig. 10, 12, figures S1 to S20

Done as suggested.

**1.** 15 'selected': explain (here or in §1 or §2 if there it takes more than a few words) selected based on what**

We mentioned in the next sentence: "*along nine profiles crossing monitored rock walls in Norway*", so it should be explicit enough that we select rock walls, where we have measurements. See also comment above.

**1.** 80 'selected sites', 'other sites' summarize how they chose those sites, as the choice can introduce biases in the derived works.**

See comment above about the selection of the sites, as being either monitored by NVE or periodically measured by NVE and NGU.

**1. 87** the model uses forcing calibrated with observations but the model is not 'observationconstrained' because no subsurface measurements are used to constrain the 2D model results.**

We changed "observation-constrained modelling" to "observation-constrained model for ground surface temperature (GST)".

**l. 91-93** this hints to the aims of the work but they need to be more clearly and explicitly stated.**

We rewrote the last paragraph in the introduction to be more explicit about the aims:

"The aim of this study is to improve knowledge about the spatio-temporal variations in ground temperature in steep rock walls in Norway on the inter-decadal scale. We employ the 2D slope-scale transient heat flow model CryoGrid 2D (Myhra et al., 2017) to simulate the thermal evolution of mountain permafrost since 1900 along nine transects crossing the instrumented rock walls in mainland Norway. We advance the methods presented in the study by Myhra et al. (2017), by an observation-constrained model for ground surface temperature (GST), i.e. including the field observations from rock walls in various expositions. All sites presented in this study are monitored by at least one rock wall logger in a vertical rock face, and three of the unstable sites are constantly monitored by the Norwegian Water and Energy Directorate (NVE). Thus, this study aims to be an important baseline for the development of the ground thermal regime in steep mountain terrain, which is possibly unstable."

**§2**

**This section needs to state clearly how the selected sites were chosen, how each transect was drawn, why none of the sites with boreholes were modelled, or if they were, why no comparison is shown between model and measurements.**

Thanks for the valid comment. As mentioned earlier in the replies we chose sites where we have measurements. We validated the model against several boreholes in the gentle terrain in the Jotunheimen Mountains. This is addressed in depth in our replies to Anonymous Referee #1, and we choose not to repeat the answer here in addition.

**Most of the place names mentioned in this section are not shown in fig. 1 nor any of the map so they aren't useful to any reader who isn't already familiar with these regions. Places that really need to be mentioned usually also deserve to be shown on a map.**

We rewrote Section 2, so there are no geographical names for smaller areas, except for the Jostedalsbreen Ice Cap. We added location of the boreholes to Figure 1D and refer to them in the text.

Consider reducing unessential info on geography, tectonostratigraphic units and mineralogy, and descriptions of slope instabilities at other sites than the modelled profiles unless they share some fundamental similarity with the latter.

Done. See the revised manuscript.

**§3.1**

If CryoGrid 2D has been compared to actual borehole measurements, cite the relevant publications, summarize the observed performance of the model, and indicate any difference in the model configuration used here. If not, mention that the model has not yet been validated. The point here is that heat conduction and numerical solvers are well understood tools, but any new implementation needs to be proven correct. Additionally, the impacts of not accounting for convective and advective heat transport, fractures (both filled or open, with or without circulation of air or water), as well as the chosen ways of accounting for surface material and snow all reduce the accuracy of the model by an unknown margin. This is especially important in landscapes prone to slope instability and close to freezing temperatures, such as many of those discussed here, where more pervasive and widening fractures are to be expected and the any infiltrating surface waters can have a large impact on stability.

We again emphasize that we do not aim to present a new model which can account e.g. for the effects of fractures, convective transport, as well as complex water flow within the rock massif. In fact, we are not aware of any model that could deliver such simulations at the spatial and temporal scales of this study. In the model description, we have made it clear that CryoGrid2D is a purely heat-conduction-based model, and we openly discuss the limitations of its application in the Discussion. Furthermore, modelling of non-conductive heat transfer mechanisms would require good knowledge about fracture geometry in the subsurface, which we do not have. In addition, coupling the conductive model with other heat transfer mechanisms on longer time scales would not necessarily yield more reliable results due to increasing model complexity and even more input data would be required on longer time scales.

Regarding validation, we compared modelled ground temperature along the Galdhøe profile to borehole measurements to validate the model. We added "Supplementary Figure S24. Comparison of annual modelled ground temperature at depths below 5 m along Galdhøe profile to measured ground temperatures by 100 m deep PACE borehole in the Jotunheimen Mountains. RMSE-root mean square error, MAE-mean absolute error, ME-mean error. Refer to Figure 1 in the main manuscript for the borehole location.", which shows R2=0.75, RMSE=0.08, MAE=0.06, ME=0.03. Other boreholes in the Jotunheimen Mountain Range are shallower, therefore we only compare modelled and simulated ground temperature at 10 m depth for all boreholes over the 2010s ("Table S5. Comparison of simulated ground temperatures at the Galdhøe profile and measured ground temperatures in the Jotunheimen Mountains over the 2010s. Refer to Figure 1 in the main manuscript for the borehole locations"). The latter table includes three snow scenarios ("Main", "nF+0.1", "nF-0.1") and the absolute difference between modelled and measured ground temperature is maximum 0.52 °C. The compared period length and depths are justified by the study focus on the inter-decadal variations in the ground temperature.

Etzelmüller et al. (2022) also showed that, the larger spatial distribution of ground temperatures is reproduced by CryoGrid 2D.

**§3.2**

State how the exact trace of the profiles was drawn and how its geometric relation with the topography may influence the results. Fig. 1 seems to show that many of the chosen profiles have long sections running at a slant angle from the slope, e.g. the first half of Hogrenningsnibba and most of the Kvernhusfjellet profiles. The Mannen profile between 1000 and 1500 m even follows the top of a ridge. While some compromise is unavoidable when dealing with real topography, these profiles are very far from the requirement of a 2D model that lateral heat fluxes though the plane of the model are zero.

We agree with Anonymous Referee #2, but the simplification to 2D slices is made in many similar studies, simply due to limits on computation. While there certainly are 3D effects, our transects represent the overall elevation and slope/aspect characteristics of the sites, in line with the overall goals of the study. If for instance the Mannen profile was moved to either side, the profiles would certainly not look entirely different. In the revised version, we have openly mentioned this limitation in Subsection 5.1.1:

"We note that our transects are only approximately suitable for two-dimensional heat conduction; yet they still follow the general characteristics of the slope and are representative of their surroundings."

**§3.3.1**

**1. 239-240 unclear whether the trend is with elevation or with time.**

Both, since SAT trends are both overestimated with elevation and with time at higher elevations. These two are connected. The SAT trends are probably not overestimated in the valleys; however, they are less important in our study. We believe this sentence should be clear:

"However, the seNorge data set overestimates SAT trends and often shows positive SAT trends with elevation for our study sites, leading to e.g. 3 °C SAT increase in Jotunheimen between the 1980s and 2010s." We mention the issues with SAT trends with elevation, which result in the issues with SAT trends with time at higher elevations between the 1980s and 2010s.

**l. 246 'coverage' -> 'resolution' (?) Also state what the resolution is.**

The spatial resolution is 2x2 km, so it is a bit less than the seNorge data sets (1x1 km). However, the resolution of the data itself is not a problem. The problem is that the data from Hanssen-Bauer et al. (2006) gives good temporal estimates at a regional scale; however, it does not resolve local scales so well. We changed the sentence to:

"This regional model yields robust temporal estimates at a regional scale; however, the data provides rather poor temperature series at local scales."

We added the spatial resolution to this sentence:

"Therefore, we choose to force the model with the regional monthly data set at 2 km spatial resolution provided by the Norwegian Meteorological Institute, described in detail in Hanssen-Bauer et al. (2006)."

**l.** 252 (step 2) doesn't this bring back the problem that the nearby valley bottom meteo station or seNorge grid cell overestimate cold periods?**

The mentioned offset is a mean value over the last few years when there was better data coverage. Additionally, we do not compute any air temperatures for higher elevations in this step, so it does not really matter. The mentioned issues with the seNorge data set are the biggest problem at high elevations due to the poor spatial data coverage.

**1. 254-255 (step 3) this lapse rate will be affected by inversions, is this intentional?**

Yes, it is intentional. The lapse rate includes air temperature inversions and is typically smaller during winter. Otherwise, air temperature at the mountain plateaus would be far too low. The most important difference between our data and the seNorge is that we use the lapse rates from the period when there was more meteorological stations and more data from the mountain plateaus, even in the seNorge data.

**§3.3.2**

**The method used to make up for the lack of snow cover observations is reasonable, but it is quite simplified and uses many arbitrarily set values (Table 3) so again the model performance will be affected in an unknown way without borehole measurements.**

We have references for some values in Table 1. We are not aware of any other studies we could use to validate our values. We discussed issues with snow distribution and the thermal effects of snow cover in Subsections 5.1.3 and 5.1.4. We added model validation for three snow scenarios in Supplementary Table S5, and the largest difference for all scenarios and borehole is maximum 1.30 °C, showing that there is a good agreement with the boreholes even for the less suitable snow scenarios.

**§3.4**

**1. 302 'correlation'?**

We removed "*for correlation*". This was relevant for the previous model runs, where the air temperature data series was generated using correlation; however, we changed it in the current version as defined in step (1) in Subsection 3.3.1.

**l. 318-320** Explain what is an uncertainty run, what a control run (never mentioned anywhere else in the text) and what are the uncertainty simulations mentioned in the text but not here. Are 'run', 'simulation' and 'scenario' synonyms throughout the text?**

"Run", "simulation" and "scenario" are synonyms in our manuscript.

We decided to harmonize the terms in the revised version to be clearer:

- 1) We only use word "simulation" and replace all "run/scenario" with "simulation".
- 2) We call all simulations for "*simulations/sensitivity simulations*". We change "*control runs*" to "*test simulations*". We modify the text accordingly:

"We evaluate model sensitivity for all profiles by rerunning the model, including the initialisation steps. However, we note that some simulations are to check the thermal influence of likely uncertainties in the model forcing or parameters ("uncertainty")

simulations"), and some are "test simulations" to investigate the thermal influence of e.g. nearby glaciers, sediments or SOs in the rock walls. Uncertainty and test simulations are listed in Table 3."

We also inserted a column with "*simulation type*" in Table 3, where we specify uncertainty and test simulations.

**§4.1**

**1.326-328** 'observations-constrained modelling...': it's more clear for the reader to call this 'calibration of GST forcing input using the measured SOs', similar to how it is already worded later on at 1. 696. Once calibrated, of course the mean error is 0.00 for all 20 loggers, so the captions of figures S1 to S20 calling it 'validation' are misleading since the comparison and statistics are done against the same data used for calibration.

We changed "*observations-constrained modelling*..." as suggested. We did not divide our data into validation and calibration data sets, so we do agree it should be called calibration and not validation. We changed the figure captions.

**§§4.2-4.3-4.4 (l. 329-557)**

These sections are much too long compared to what the reader gains from them. It's not necessary to describe in words everything that is visible in the figures, unless some feature contradicts other evidence or solves an open questions about some important point along a transect. Commenting each scenario and the details of each site is quite uninteresting because the model hasn't been validated using borehole data neither at these sites nor elsewhere (ortherwise provide reference). So the model outputs remain speculative and cannot be used to draw such detailed conclusions as described here about any one specific site.

I suggest these three sections are shortened and rewritten so that instead of discussing in unwarranted detail the state of each site under each scenario, they discuss how similar profile features (e.g. shape of the slope, presence of blockfields) and model scenario across different sites may lead (or not) to similar model outputs. This would show whether the model behaves consistently under similar conditions. Some of this is currently in §5.3. If instead your primary focus is on describing the conditions at each site, consider consolidating all info on each site in one subsection, discuss primarily the 'main' scenario and mention the other scenarios when they clearly improve the understanding of the 'main' scenario.

We thank Anonymous Referee #2 for this comment and suggestion. We agree that it was maybe too long. We have rewritten the section as suggested and changed the associated figures. Some of the figures are moved to Supplementary Figures or removed. See the revised manuscript.

**§5.1.1**

Four major limitations are not mentioned:

- the correctness of the CryoGrid 2D code has not been validated against observations either in this study or in the literature. Magnin et al. 2017 notes that validation against boreholes measurements is rare due to lack of data and because established heat conduction models are assumed to be reliable under simple thermal systems. But CryoGrid 2D is not yet an established code. Myhra et al., 2017 didn't have supsurface measurements, Kristensen et al. 2021 applied it to an unstable slope without any validation dataset, and in a setting that is clearly not a simple thermal system based on the extreme heterogeneity of the unstable volume shown in their fig. 5 and on their finding that permeability varies with temperature.

We validated the model for the gentle terrain. More details in our general replies to Anonymous Referee #1.

- most of the sites chosen in this study are close to unstable slopes where fractures can be expected to be particularly frequent, pervasive and quite possibly open or filled with ice or water, i.e. not at all a simple thermal system that can be modelled with confidence by assuming heat conduction dominates below just a few metres.

We agree with Anonymous Referee #2 that there may be fractures which can at least locally modify the thermal regime. However, it would require immense resources to map the fracture structure and evolution) at the spatial and temporal scales targeted in this study. We are not aware of any study that has attempted this. Without knowledge on fracture distribution, however, it is better to restrain modelling efforts to the key processes, and heat conduction through the bedrock is clearly the most important. We have added a discussion on the impact of ice/water-filled fractures in the Discussion under limitations:

**"If ice/water-filled fractures inside the bedrock exist, this would locally delay permafrost thawing/formation due to latent heat effects."**

- the chosen profiles, especially (but not only) at Hogrenningsnibba, Kvernhusfjellet and Mannen have long sections where the geometric relationship to the slope (e.g. crossing it at a slant angle, or running along a ridge) is such that significant lateral heat flux through must be expected, violating a basic assumption of 2D models. Clearly this type of profiles and topography are not what Myhra et al., 2017 referred to when arguing for the suitability of 2D models in the Norwegian mountains because of their flat plateaus and long valleys.

We agree with Anonymous Referee #2, but the simplification to 2D slices is made in many similar studies, simply due to limits on computation. While there certainly are 3D effects, our transects represent the overall elevation and slope/aspect characteristics of the sites, in line with the overall goals of the study. If for instance the Mannen profile was moved to either side, the profiles would certainly not look entirely different. In the revised version, we have openly mentioned this limitation in Subsection 5.1.1:

"We note that our transects are only approximately suitable for two-dimensional heat conduction; yet they still follow the general characteristics of the slope and are representative of their surroundings."

**§5.1.2**

**The use SOs calculated at each logger from GST measurements is mentioned in many places as a key strength of the study but where the SOs from different loggers are used is not well explained except for table 4.**

We added Supplementary Figure S1 with a caption: "Supplementary Figure S1. Logger data used for the computation of surface offsets arising from the incoming solar shortwave radiation along profiles. Suffix "l"=at a lower elevation, suffix "h"=at a higher elevation.", where we clearly show where and which loggers we use to account for the surface offsets in the steep rock walls. Furthermore, we added a sentence in Subsection 3.3.3.: "Supplementary Figure S1 shows more details about the applied loggers along profiles."

**Each transect runs across a range of elevation, slope and aspect widely deviating from those of the closest logger.**

Yes, that may be the case. We use the data from a logger, which we assumed was most representative for the main aspect of a rock wall, which is not necessarily the closest logger. We believe it makes more sense to use data from a more representative logger than from the closest logger.

**It seems that most transect 'main runs' use different loggers for their easternmost and westernmost slopes, is this so?**

Yes. Check Supplementary Figure S1 for details.

Then some are run with alternative loggers too. Calling these 'sensitivity scenarios' is confusing because they are not investigating the sensitivity to some pameter of the model configuration, but rather creating alternative forcing scenarios based on less realistic calculated SOs.

For some profiles the alternative runs are just "*test simulations*", so they are not included in our "*uncertainty simulations*" and the GT range shown in Figures 4,5,6 in the previous manuscript version. Note that the term "*sensitivity simulations*" includes both "*uncertainty simulations*" and "*test simulations*". We updated Table 3 with the details about our assumptions. However, this is less important in the current manuscript version, since we removed figures with the GT range to shorten Section 4.

Magnin et al. (2017) remarks that accurate GST are a prerequisite for a scheme without surface energy balance and without snow. Givn the mentioned frequent occurrence of temperature inversions, if GST is only calibrated at one logger along a transect (or one per mountain side) and than used everywhere along that transect or mountain side, accurate GST can't be assumed and the reference to the good performance below 6-8 m of the Magnin et al. (2007) model is unwarranted. See also my note to figg. S1-S20

It seems our modelling steps were not so clearly explained. We do not use the same GST along a mountainside and air temperature inversions are included in our SAT data by using different monthly lapse rates, which are typically lower during the winter months. To be clearer about our choices, we changed step (4) in Subsection 3.3.1:

"(4) We compute monthly SAT along the profiles using monthly lapse rates."

**and add a sentence at the end of Subsection 3.3.1:**

"After generation of the SAT data sets, we account for the nival offsets and surface offsets arising from the shortwave solar radiation (See Subsections 3.3.2 and 3.3.3.) by modifying SAT along the profiles."

**The last part should be clear:**

"Instead of using temperature transfer factors, we add measured average monthly SOs to SATs at the location of rock walls along profiles."

as mentioned in Subsection 3.3.3. See also our reply to a similar comment under §6.

**§§5.1.3, 5.1.4**

Because of the very effects described in these sections, snow is another element that makes it impossible to rely on the performance of the model without validation against borehole measurements. Especially when most of the nF values in Table 3 are rather arbitrary round numbers.

We validated the model performance, where it was possible. We addressed it in our general replies to Anonymous Referee #1.

The claim that snow effects are in some measure also accounted for through the use of mean monthly SOs computed from GST measurements is significantly weakened if only one logger per scenario is used along each entire transect, or side of the mountain, as discussed in my comment to §5.1.2.

We addressed in the comment to Section §5.1.2 that it is not one logger used per scenario or mountainside; however, SOs from one logger are used along the steepest part for various rock wall aspects. Some of the rock walls are rougher, and it is likely they have more snow during winter than smooth rock walls, e.g. the north-facing rock wall at Gámanjunni is quite rough, and hence it makes sense that it can accumulate snow. See also our reply to a similar comment under §6.

**§5.2**

Is this section's primary focus on finding whether GST is a reliable indicator of permafrost by comparing where GST < 0 vs. where the 2D model predicts permafrost? Or on comparing the 2D model predictions vs. other studies that used SAT or GST predicted permafrost? Or is the aim is to suggest what the most likely conditions at each site are?

I recommend focusing on a brief and clear comparison between the permafrost/no permafrost prediction of the model match earlier published studies.

We removed any information about rock wall loggers from Section 5.2. We decided to move the information about the measured GSTs to "4.1. Surface offsets and logger data" and we added "Supplementary Table S4. Logger data. Mean ground surface temperature (GST) is computed as a mean for the mentioned periods." The following text is added to Section 4.1:

"Supplementary Table S4 includes information about the measured GSTs at the study sites. Mean rock wall temperature at or below 0 °C over at least two consecutive years usually indicates permafrost; however due to lateral heat fluxes and the preservation of long-term temperature signals at depth, permafrost may occur even if rock wall logger temperature is above 0 °C (Noetzli et al., 2007; Noetzli and Gruber, 2009). All loggers at Mannen and the Wfacing logger at Ramnanosi suggest an unlikeliness of permafrost presence in these rock wall expositions over the last few years. The north-facing logger at Ramnanosi measured mean rock wall temperature at 0.02 °C (Aug 2016–Jul 2020; 1370 m); hence, permafrost was likely in the north-facing parts of the slope, at least before the measurement period started. The northfacing logger in the Loen area indicates that permafrost is likely, whereas the west- and southfacing loggers have positive temperatures. In Jotunheimen, most rock wall loggers indicate that even cold permafrost exists in the Jotunheimen Mountains. In the Gámanjunni area, at least warm permafrost conditions can be expected in the rock walls. For Ádjit both loggers indicate permafrost, although the south-facing rock wall is close to non-permafrost conditions. All loggers at Rombakstøtta, except for the east-facing logger, indicate that at least warm permafrost may be present in the rock walls."

Furthermore, we moved this sentence to Subsection 5.1.1:

"Since some cracks exist on the plateau above Mannen (Saintot et al., 2012) and Ramnanosi, air ventilation could lower GT in the area; however, since thick snow cover accumulates on the Mannen Plateau, plugging of the cracks with snow could prevent air ventilation (e.g. Blikra and Christensen, 2014)."

We renamed Section 5.2. to "5.2 *Comparison to borehole data, geophysical surveys and other studies*" and kept only comparison to borehole data, geophysical surveys and the following studies about permafrost in Norway: Isaksen et al. (2011), Blikra and Christensen (2014), Frauenfelder et al. (2018), Etzelmüller et al. (2022). See the revised manuscript for details.

1. 733-735 Given that the major weakness of the ms. is that the 2D model is unproven, mentioning the existence of data from some boreholes in Jotunheimen that are suitable for comparison without showing such comparison is disconcerting. Claiming they 'agree quite well' is unwarranted. The comparison with borehole data needs to be shown and documented, especially as the boreholes are said to span different snow conditions. If this is not possible, the existence of the data needs to be mentioned for future reference but without making unsupported claims of good model agreement.

This is addressed in our general replies to Anonymous Referee #1.

**§5.3**

This is by far the most valuable part of all §5 and in fact the section of the manuscript that I feel provides most insight, even considering the limitations of the underlying model. It may be expanded a bit after other sections are shortened.

We added two sentences to this section.

**§6**

These conclusions need to be qualified by stating that the model used has not been validated at depth and that it cannot account for effects that are especially important in landscapes prone to slope instability, such as many of those discussed here, where more pervasive and possibly open fractures is to be expected. Also mention (maybe in the Introduction) that a heavily fractured rock mass is thermally different and can be expected to have deeper reaching impacts on temperature compared to the local and shallow effect of a single fracture some metres below the rock face, like the one described in Magnin et al., 2005.

We have addressed a similar comment above.

**l. 812-814** this contradicts the claims in the earlier sections that the model results below 6 m are insensitive to the details of how snow is treated.**

This is not necessarily a contradiction, because:

(1) Validation of any model (here the model from Magnin et al, 2017a) can show that the modelling results agree quite well with the measurements, even though some smaller thermal effects are unaccounted for. GT can be influenced or vary in space; however, the question is whether is varies so much that the model validation would show that the agreement is insufficient.

(2) The snow conditions influence GT distribution particularly in smaller rock walls (i.e. with ledges both above and below them), which were not considered in the study of Magnin et al. (2017a). In addition, Magnin et al. (2017a) do not consider any rock walls below mountain plateaus, where large amounts of snow could accumulate on the plateaus. We decided to modify the following sentence to emphasise that our reference is mostly about high rock walls and sharp peaks:

"Magnin et al. (2017a) showed that a similar approach, i.e. without energy balance and without consideration of snow accumulation in rock walls, was appropriate to reproduce temperature below steep flanks of sharp mountain peaks at depths > 6 or > 8 m by comparing the modelled temperature to the measured temperature profiles in boreholes."

**l. 821** this confirm that the model isn't necessarily accurate accurate below a few metres if only one SOs calculated from one GST measurement is used over an entire transect or mountain side for each scenario.**

We have limited modelling choices, since we only have one logger available for chosen aspect for each site. However, note the similarity between seasonality in SOs at Ádjit and Gámanjunni (e.g. Figure 3 and the figures below). Even though these two sites lie a few km from each other, they have quite similar seasonality in SOs for the south-facing and north-facing loggers, confirming that our assumptions are not necessarily wrong. We note that the two sites have similar bedrock. The minor differences in SO seasonality between the loggers could be due to the issues with lapse rates at the sites. We add figures below, which show the high correlation between GST measured by the loggers and moderate to high correlation between their SOs.

---

## Author Response (AR2)

**tc-2022-4**

**Post Little Ice Age rock wall permafrost evolution in Norway**

Justyna Czekirda, Bernd Etzelmüller, Sebastian Westermann, Ketil Isaksen, Florence Magnin

**General response to referees**

We would like to thank the two referees for their reviews. We modified the manuscript according to the comments made by the referees.

**Response to individual referees**

**Referee comments are in bold**
Author replies are in normal text (*Italic is for the manuscript text*)

**General Comments**

**Czekirda et al. have submitted a revised manuscript in response to reviewer comments. A detailed response addressing the major comments has been provided. This is much appreciated. The authors appear to have addressed the main concerns that I raised, and the manuscript is much improved. In particular, the authors have included a more detailed comparison between the simulated ground thermal regime and observational data (ground temperatures and geophysical surveys) available for the region. Although deeper temperature data and geophysical surveys in the region are limited (particularly in close proximity to the study transects), the authors have made good use of available data to support validation of simulated results. This strengthens the paper and increases confidence in the conclusions regarding the evolution of rock wall permafrost in Norway. The clarifications and additional information provided in the revised MS are also appreciated and these will make the paper easier to read especially for people that are less familiar with permafrost conditions in Norway including mountainous regions.**

**I have no major concerns with the revised MS. However, I do have some minor comments, many of which are editorial and are meant to improve language. As mentioned in my previous review, this is an interesting study and with some additional minor revisions it should be acceptable for publication**.

We want to thank Anonymous Referee #1 for the additional suggestions how to improve the language and correct our language pitfalls.

**Additional Comments**
**L13, L14, L16, L17-18, L19, L20, L21, L22, L26, L30-32, L42, L46, L53, L82, L83-84, L85-86, L92, L92-93, L96, L97-98, L98-99, L111, L112, L120-121, L125, L131-132, L133, L138, L163, L262, L269, L270, L277, L278, L280, L286-287, L288, L295, L301, L302-304, L332, L340, L364, L388 (typo? We corrected L358), L390, L405, L511, L548, L563, L592-595, L623-627, L677 (typo? We corrected L657), L678 (typo? We corrected L658), L669, L694, L702-703, L706, L714-715, L718, L719, L720, L764, Figure 4, 6, 7, 8**

Done as suggested.

**L72-73 – In the earlier review, I mentioned that the type of sensor and its accuracy and precision are usually provided in the methodology section. I realize that you have included it here but normally this level of detail isn't provided in the Introduction (i.e. we would just say that "field observations of rock wall temperatures were acquired at selected sites in previous studies") and the details on instrumentation utilized would be provided in the methodology section.**

We removed this information from Introduction, and moved it to each subsection in Section 2 Study areas and field installations:

> 2.1 Western Norway: "*During 2015–2017 nine Geoprecision, M-Log 5W Rock loggers with at least 0.1 °C at 0 °C accuracy were installed at selected rock walls to measure surface temperature in Western Norway (Magnin et al., 2019).*"

2.2 The Jotunheimen Mountains: "*Furthermore, Geoprecision, M-Log 5W Rock loggers (at least 0.1 °C at 0 °C accuracy) were installed at selected sites in Jotunheimen (Hipp et al., 2014).*"

2.3 Northern Norway: "*All sites are instrumented with Geoprecision, M-Log 5W Rock loggers with at least 0.1 °C at 0 °C accuracy.*"

**L75-76 – Do you mean that the data acquired from these earlier studies helped to calibrate the model?**

Yes, we modified it: "*From 2015 through 2017 other sites across Southern and Northern Norway were also logged (Magnin et al., 2019), allowing for the improvement of earlier approaches by Hipp et al. (2014) and Steiger et al. (2016). The acquired data helped to calibrate a near-surface thermal regime model for rock wall permafrost in Norway, by using mean annual air temperature (MAAT) as an explanatory variable instead of elevation.*"

**L158 – specify whether the resolution given is vertical or horizontal**

We skip this information since we believe it is generally accepted that resolution of a 2.5-dimensional DEM represents a horizontal grid cell, and vertical resolution would be probably better described as e.g. its vertical accuracy, which we do not mention at all.

**L191-192 – unclear – do you mean bedrock is at the ground surface?**

No, we mean that we use bedrock class to assign volumetric contents of the soil constituents from the interface between glaciers/perennial snow and ground below all the way down to the lower model boundary. We rewrote "*Bedrock stratigraphy is assumed to be below glaciers and perennial snow.*" To "*Bedrock class (Class "a" in Supplementary Table S1) is assumed below glaciers and perennial snow*".

**L250-253 – The issue is probably that nT and nF together consider surface offset.**

nT and nF-factors mostly influence GST in the months when surface air temperature has largest magnitudes. During spring months, surface air temperature usually has small magnitudes regardless of sign and neither nF nor nT can account for the large surface offsets in rock walls during spring months.

**L312 and L316 – Sporadic permafrost is also discontinuous. "sporadic to extensive discontinuous permafrost" or "sporadic to widespread permafrost" is probably what you mean**

It depends on the definition of permafrost zones. We are aware of the mentioned definitions; however, permafrost zones are usually as follows: continuous (90-100 % area), discontinuous (50-90 %), sporadic (10-50 %), and isolated (0-10 %). For instance, the most recent global permafrost map uses this definition (Obu et al, 2019; doi: 10.1016/j.earscirev.2019.04.023). So, to be specific, we added in the first sentence on L312: "*indicate that sporadic (10-50 % area) to discontinuous (50-90 % area) permafrost*".

**L319 – are you referring to GT or GST here?**

We refer to GT field, although note that it also includes GST at the surface. Hence, we do not really see the need to specify whether we mean GT or GST for the two-dimensional case in our study.

**L574-573 – I assume in this section you are referring more to spatial variation rather than temporal variation (i.e more a comparison between sites with more or less snow cover). In your last sentence, you could consider saying that your measurements don't indicate a negative SO.**

We suggest to keep this part as it is since we mentioned in the previous sentence that "*the overall annual cooling of the ground surface due to snow cover is not observed in Norway*".

**L604 – Reduces convective cooling?**

We keep this part, because this is how the authors describe the process in the cited paper.

**L730-733 – This bit about "hot spots" is unclear.**

We removed "hot spots" from the sentence and modified it: "*Thus, especially the constant change of ground thermal regime associated with rock walls and their vicinity facilitates material production and further geomorphological transport processes.*"

**Figure 2 – Be clear about what that red and green sections are.**

We added in the caption: "*Different colours near the surface show various stratigraphic layers (See Supplement Table S1 for details)*".

**Figure S24 – Are you considering temperatures measured from 5 – 100m depth. You refer to 100m borehole but not clear if you are comparing temperatures to this depth.**

Yes, we do compare to temperatures measured at depths between 5 to 100 m. We modified the caption: "*Supplementary Figure S24. Comparison of annual modelled ground temperature at depths between 5 and 100 m along the Galdhøe profile to ground temperatures measured in PACE borehole in the Jotunheimen Mountains. RMSE-root mean square error, MAE-mean absolute error, ME-mean error. Refer to Figure 1 in the main text for the borehole location.*"

**Anonymous Referee #3**

In their manuscript, Czekirda et al. model the evolution of rock wall permafrost for selected profiles in Norway since the Little Ice Age using the CryoGrid2D model. They use nine transects for their study that cross steep rock walls at high elevation, where rock wall temperatures have been monitored, and assess temperature changes over the past 120

**years. Their results indicate rising ground temperatures, especially following the 1980s. Moreover, they find that within individual rock walls there is a trend in warming rate with elevation. Though the model is rather simplistic (2D case, only conductive heat transfer), the authors gain interesting insights into e.g. the role of snow cover and the steepness of rock walls, and discuss these parameters in great detail. The paper is potentially of interest for a large readership, especially those working on slope (in-)stability and permafrost distribution in mountain areas. This is obviously a revised version of the original submission that was commented on in great detail by two anonymous reviewers. After thoroughly addressing all concerns raised, rephrasing large parts of their manuscript and adding a new section on the geomorphological implications of the study, the paper significantly improved in quality. It is now very well written and merits publication in TC. I recommend to accept the manuscript after addressing a few technical corrections mentioned below.**

We thank Anonymous Referee #3 for his/her comments, and we try to address the specific comments in the following.

**Specific comments**

**L46, L85, L425-426, L657**

Done as suggested.

**- L230: I suggest to quickly explain the idea and values of the "freezing n-factors" here in a brief sentence**

We modified and added sentences about the nF-factors: "*In equilibrium permafrost models such as the TTOP-model (Smith and Riseborough, 2002), insulating snow effects are usually accounted for by using semi-empirical transfer functions, so-called freezing n-factors (nF). The nF-factors link SATs and GSTs by relating the freezing degree days at the surface to the air. In Norway, the freezing n-factors vary between 0.1 for the attenuation effects of deep snow cover to 1.0 for very thin or absent snow cover (Gisnås et al., 2013).*"

Gisnås, K., Etzelmüller, B., Farbrot, H., Schuler, T. V., and Westermann, S.: CryoGRID 1.0: Permafrost Distribution in Norway estimated by a Spatial Numerical Model, Permafrost Periglac, 24, 2-19, https://doi.org/10.1002/ppp.1765, 2013.

**- L228-239: Is there some factor included for snow redistribution towards the lower portions of steeply inclined bedrock walls?**

No, avalanche patterns are highly variable in space and time, so accounting for this in the model is not possible at this stage. We added a short sentence: "*Note that snow redistribution towards the lower portion of the slope is not considered.*"

**- L503: may occur? I would think that 3D heat transfer is the default…**

We removed "may". It is now: "*heat transfer processes in complex terrain occur three-dimensionally*".